# The feeding microstructure of male and female mice

**Yakshkumar Dilipbhai Rathod, Mauricio Di Fulvio** [ORCID] *

Department of Pharmacology and Toxicology, School of Medicine, Wright State University, Dayton, OH, United States of America

* Mauricio.DiFulvio@wright.edu

## Abstract

The feeding pattern and control of energy intake in mice housed in groups are poorly understood. Here, we determined and quantified the normal feeding microstructure of social male and female mice of the C57BL/6J genetic background fed a chow diet. Mice at 10w, 20w and 30w of age showed the expected increase in lean and fat mass, being the latter more pronounced and variable in males than in females. Under *ad libitum* conditions, 20w and 30w old females housed in groups showed significantly increased daily energy intake when adjusted to body weight relative to age-matched males. This was the combined result of small increases in energy intake during the nocturnal and diurnal photoperiods of the day without major changes in the circadian pattern of energy intake or spontaneous ambulatory activity. The analysis of the feeding microstructure suggests sex- and age-related contributions of meal size, meal frequency and intermeal interval to the control of energy intake under stable energy balance, but not under negative energy balance imposed by prolonged fasting. During the night, 10-20w old females ate less frequently bigger meals and spent more time eating them resulting in reduced net energy intake relative to age-matched males. In addition, male and female mice at all ages tested significantly shortened the intermeal interval during the first hours of re-feeding in response to fasting without affecting meal size. Further, 20-30w old males lengthened their intermeal interval as re-feeding time increased to reach fed-levels faster than age-matched females. Collectively, our results suggest that the physiological mechanisms controlling meal size (satiation) and the non-eating time spent between meals (satiety) during stable or negative energy balance are regulated in a sex- and age-dependent manner in social mice.

## Introduction

Small chronic imbalances between energy intake and energy expenditure leads to overweight and obesity [1], two known risk factors implicated in the development of metabolic syndrome and type-2 diabetes. In addition, alterations in the feeding pattern of humans and animal models have been proposed to play significant roles in the long-term regulation of adiposity and body weight (BW) [2–4]. However, whether the development of obesity is consequence of changes in meal size and/or meal frequency remain largely unknown [5–7]. In fact, it was long

**Data Availability Statement:** All relevant data are within the manuscript and its Supporting Information files.

**Funding:** This research has been supported in part by funds from the American Diabetes Association and National Institutes of Health Grants #1-17-IBS-

258 and R21DK113446-01 to MDiF. The funders had no role in study design, data collection and analysis, decision to publish, or preparation of the manuscript.

**Competing interests:** The authors have declared that no competing interests exist.

thought that frequent consumption of small meals decreases obesity risk [2, 8]. Yet, increased eating frequency by itself appears to contribute minimally to the short- or long-term control of energy intake and obesity in some settings and research models [9–11]. It has been proposed that mice consuming less frequent calories in a magnitude comparable to that of *ad libitum* are protected against obesity, hyperinsulinemia and hepatic steatosis [6, 12, 13]. Although the physiological role of meal frequency on adiposity remain uncertain [14], increasing the caloric content of a meal does increase energy intake in adult humans [5, 15–18], thus contributing to the development of obesity, if that effect was to persist in subsequent meals [19–21]. In fact, increased meal size, rather than net daily caloric intake correlated with obesity in rats [22] and in humans [23]. These results imply that obesity in response to increased meal size and energy intake may not elicit expected compensatory reductions in meal frequency [24]. The absence of compensatory mechanisms aimed at limiting energy intake by reducing meal frequency may reflect impaired feedback signals controlling satiation *i.e.*, the termination of a meal and/ or the interval of time spent not eating *i.e.*, satiety [25]. In agreement, genetically obese mice (*ob/ob*) of both sexes consumed larger liquid meals separated by apparently longer non-eating periods or intermeal intervals (IMI) [26] thus suggesting that obesity may interfere with physiological mechanisms involved in the control of satiation and satiety.

Therefore, the elucidation of metabolic, neuro/endocrine and behavioral factors involved in the termination of a meal or in the time spent not eating as integral components of energy homeostasis becomes critical to understand the mechanisms involved in the long-term regulation of BW and adiposity [27, 28]. Meal size and other parameters of the feeding pattern including meal frequency and IMI are highly dynamic variables that can be phenotypically measured in animal models and be highly informative surrogates of satiation and satiety [29–32]. However, the accurate determination of the behavioral unit of energy intake *i.e.*, meal size and consequently the analysis of the feeding pattern in humans and rodent models produced conflicting results over the years in part due to discrepant definitions used to delineate the feeding structure [9, 31, 33]. In the case of animal models, for instance, the minimum IMI or threshold (IMI$t$) necessary to define a meal in rats was empirically determined and widely different between studies [34–37], or even omitted or not considered in others [22]. Similarly, normal and mutant male or female mice models of different genetic backgrounds and ages were used to determine meal size under several conditions using empirically [26, 38] or mathematically [39, 40] defined IMI$t$. However, in some instances, no IMI$t$ criterion was used to define meals in mice [41–45] thus making the analysis of the feeding structure and the interpretation of the regulation of energy intake unreliable.

Therefore, the lack of uniform criteria to determine meal size in animal models of obesity has complicated the interpretation of the roles that satiation and satiety have in the control of energy intake. More recently, however, Zorrilla *et al.* provided for the first time a detailed analysis of the feeding pattern in the adult male rat by validating objective and mathematical criteria for meal definitions [31, 32, 46]. The IMI$t$ determined following those criteria reliably and consistently defined meals separated by a non-eating period of time with initially low probability to engage in a new meal, but increasing as a function of the non-eating time to reach a maximum while approaching the next meal, thus objectively fulfilling the definition of satiety [31, 47]. This approach to determine the meal pattern in single-housed rats was recently applied to and validated in single-housed adult male mice [30]. Therefore, these previous works provided a reliable framework to consistently assess and quantify feeding behavior and energy intake in animal models [46]. However, it remains unknown if the normal feeding behavior and microstructure of mice is influenced by sex, age or social housing. These distinctions are relevant; satiation and satiety determine energy intake in the short- and in the long-terms and energy intake is closely related to BW regulation, which changes in a sex and age-dependent manner

in mice. In addition, it has been documented that net BW in mice decreases when the number of animals housed increases [48–50]. However, some studies did not find such differences when mice where housed in groups [51–53] whereas others found it increased [54, 55] or reduced [56]. These housing-related discrepancies in net BW observed in mice of the C57BL/6J background have been attributed to several potential factors (*reviewed extensively in* [57]) including age/sex-dependent differences in energy metabolism and food/energy intake [58]. In that regard, single-housing significantly increased fat mass, cumulative energy intake and expenditure in males of the C57BL/6J background relative to pair-housed mice [53]. Similarly, energy intake appears related to changes in BW and/or composition, which in turn are sex-dependent variables [59, 60].

Therefore, based on the previous considerations it is reasonable to propose the existence of sex- and age-related differences in the feeding pattern of mice. To gain insights into the behavioral regulation of the feeding pattern in social mice, we adopted the criteria validated in rats [31] and mice [30] to describe for the first time the contribution of satiation (meal size) and satiety (IMI) in the long-term control of energy intake in female and male mice at 10w, 20w and 30w of age fed *ad libitum* a chow diet and their feeding-related responses to prolonged food deprivation. The results presented here confirm and extend previous data and provide a new framework and context on which hypotheses related to the regulation of satiation and satiety in the control of energy intake of social mice can be systematically tested.

## Materials and methods

### Animals and housing

The Animal Care and Use Committee of Wright State University approved all methods involving mice, which were carried out in accordance to relevant guidelines and regulations. All mice were on the C57BL/6J genetic background and obtained from the Jackson Laboratory (Bar Harbor, ME). Mice were bred for no more than 5 generations in our animal facility to produce sufficient mice for experiments. Mice were fed *ad libitum* a standard chow diet [Envigo, Teklad 22/5 Rodent Diet #8640, 3.0kCal/g (1.6kCal carbohydrates, 0.87kCal protein and 0.51kCal fat)] starting at weaning (p19-20) for ~35 weeks. Water was available at all times and dispensed by plastic bottles (500ml, Ancare Corp. Bellmore, NY). Weaned mice (*n* = 10 per sex) were housed in groups of 5 in standard cages (350mm x 220mm x 140mm = 0.01m$^2$, Lab Products Inc., Seaford, DE) and randomly regrouped ~1 week before experiments at 10w, 20w and 30w of age (S1 Fig) to allow or re-establish high-order social hierarchies and behaviors [61–65]. Mice were housed in a controlled environment at 22 ± 1°C, 30–70% humidity and on 12:12 light (~306lux)/dark cycle with lights off at 1830h (Zeitgeber 12). Standard housing cages (0.01m$^3$) were wire-topped and included translucent PVC nest boxes (Alternative Design Manufacturing & Supply. Inc. Siloam Springs, AR), two cardboard tubes (41mm x 82mm, Jonesville Paper Tube Corp. Jonesville, MI) and cotton nestlets (50mm x 50mm, Ancare Corp. Bellmore, NY) on sterile pine sawdust bedding material (Teklad, Envigo. Indianapolis, IN). Husbandry procedures were carried out weekly. Male and female littermates were studied at 10w, 20w and 30w of age in groups of 10 in the HM2 System's cage (0.1m$^3$, *see below*).

### Radiofrequency identification of mice

Mice were identified based on radio-frequency identification (RFID) transponders implanted subcutaneously. The transponders (2.12mm x 12mm, UNO-MICRO-12, Med Associates Inc. Fairfax, VT) emit a unique barcode that is interpreted by the radio-frequency (RF) sensor (reader frequency: 134.2 kHz) located near the feeders of the automatic feeding system used in

our experiments (*see next section*). Transponders were implanted under the dorsal skin of the neck and close to the head of conscious mice at 8-9w of age (S1 Fig), essentially as described [66]. RFID-implanted mice were returned to their cages immediately after implanting transponders and monitored every 2 days to assess well-being and verify identification by using a hand-held RFID monitor (PetScan RT100, Real Trace, Villebon-sur-Yvette, FR).

## Feed and water intake system

Before experiments, ~10w old RFID-implanted mice were individually weighed and identified by using a RFID-equipped scale attached to the Feed and Water intake activity monitor system (Model HM-2, MBRose, Faaborg, DK) designed for social mice (S1 Fig). All 10w old male or female mice (*n* = 10) were placed in the wire-topped cage (540cm × 400cm × 460cm = 0.1m$^3$) of the HM-2 system for a week to allow acclimation. During the first days of this period, few fighting events were observed among males. However, once the expected social hierarchies were established [65, 67], fighting episodes became infrequent, in particular when male mice were housed in a bigger cage [68], such as that of the HM-2 system. Fighting events in 20-30w old males were rare. The HM-2 testing cage (0.1m$^3$) included the same nest boxes, cardboard tubes and sheets of absorbent paper from their previous standard housing cages (0.01m$^3$). Fresh sterile sawdust bedding material was provided and husbandry procedures were carried out weekly. After 3 weeks of monitoring, ~13w mice were randomly placed back into their standard cages (5 mice/cage) for additional ~6 weeks. When mice reached ~19w of age, they were randomly scrambled again between the two standard cages and kept for an additional week in their room. When mice were ~20w of age, all of them *i.e.*, 10 mice were placed in the HM-2 cage to test their feeding behavior for 3 weeks, procedures that were repeated once more when mice reached ~30w of age (S1 Fig). The monitoring system is equipped with two narrow squared tunnels to allow the access of a single mouse at a time to two independent feeders attached to scales. Each tunnel is surveyed by infrared beams that detect tunnel activity in the form of entry and exit of a single mouse. The tunnels are also equipped with RF detector coils to record the identity of a mouse implanted with a unique RFID transponder. The HM-2 system is also furnished with motion sensors that compute total mice ambulatory activity in the whole cage and store the data as time (mins) spent lying still or moving. The HM2 system was located in a small room separated from the main hall of the *vivarium* by two doors. The mean ambient temperature and humidity of the room were 22 ± 1˚C and 30–70%, respectively, and the light cycle was controlled by using a 24h Digital Multi-Purpose Time Switch (Tork$^®$, NSi Industries, LLC, Huntersville, NC) to provide 12:12 light (~306Lux)/dark cycle with lights off at 1830h. The HM-2 apparatus does not provide light and the computer monitor was turned off during the night to minimize all sources of external light.

## Energy intake and data collection

Food intake was recorded in real time in the HM-2 system, which determines the weights of feeders containing standard chow diet (Envigo, Teklad 22/5 Rodent Diet #8640, 3.0 kCal/g) detecting "non-eating" events when the weight is stable or "eating events" when unstable (load resolution 0.001g, meal start/end detection 5s). The interactions of the mouse with the feeders is recorded as the difference in weight before and after a feeding event. In other words, a single feeding event is recorded as a vector with a start time, duration and amount consumed and therefore, a feeding event carries its size in grams and timestamps in milliseconds indicating its initiation and termination. The data obtained by the HM-2 station, including BW of individually identified mice, were stored in a HMBase SQL database (Firebird$^®$ relational database management system) and queried by using the HM2Lab software (MBRose, Faaborg, DK)

installed in an embedded computer. The records of the food monitoring system were organized as raw data in the database and analyzed by the HM-2 software or exported to spreadsheets for manual analysis. We have performed the latter to verify, validate and extend the automatic software-based analysis.

## Meal definitions and feeding pattern

Several definitions were adopted to determine the feeding microstructure of group-housed mice. For instance, a unit of food intake *i.e.*, a meal was defined as a cluster of feeding events or bouts separated by short intervals of time in turn separated from the next meal by longer intermeal intervals (IMI, see Fig 5A). The IMI threshold (IMI*t*) *i.e.*, the minimum interval of time separating two consecutive meals is here defined by following Zorrilla's criterion [31]. Briefly, meal frequency (counts per mouse), meal size (kCal) and the rate of change in meal size (Cal/min) during the nocturnal and diurnal photoperiod of the day were estimated as a function of assumed IMIs ranging from 1min to 30min (see Fig 5B–5F). IMI*t* was defined as the IMI that produces the minimum change in meal size rate [31] and their value was ≥5min. We also defined diurnal IMI*t* ≥8min and the minimum meal size of 0.050g determined from the size of single feeding events when the IMI equals zero. Since no major differences in the diurnal feeding pattern of group-housed mice were noted by using IMI*t* ≥5min or IMI*t* ≥8min (S2 Fig) and no maximal meal size was imposed, a single meal is here defined when the sum of single feeding events is ≥0.050g and are clustered within 5min of the next feeding event. The following parameters were calculated: meal frequency (the number of meals in a defined period of time), meal size (the caloric content of a single meal), meal duration (the time that a meal last), intermeal interval (the time spent not eating), time spent in meals or meal duration (the time spent eating) and the rate of ingestion or feeding rate (the ratio between meal size and its duration, *e.g.*, the number of calories consumed per second). First meal size or duration were defined as the average size or duration of the first nocturnal meal or those of fasted mice immediately after allowing re-feeding. Net energy intake reflected the product of mean meal size and its frequency, as expected. Meal pattern data were interpreted according to the definition of satiation *i.e.*, processes that terminate a meal thus reducing meal size, and of satiety *i.e.*, the non-eating time spent between meals directly modulating meal frequency independently of meal size [69].

## Fasting protocol

Mice fed *ad libitum* in the HM-2 system were not allowed to feed from 1600 to 0800 (16hs fasting) during the penultimate night of each feeding sessions (S1 Fig) to reduce their net daily energy intake by ~75%. Mice BW was recorded before fasting and immediately before allowing re-feeding to record BW drop during fasting. Net BW gain and net energy intake were then determined during re-feeding at 2, 4, 6, 8 and 24hs. During fasting, water was freely accessible and ambulatory activity monitored. During re-feeding, the eating behavior of mice was continuously recorded to determine meal size, frequency and duration, intermeal interval, feeding rate and cumulative ambulatory activity.

## Body temperature and composition

Thermometry in mice was achieved by rectal probing to facilitate the detection of fasting-induced torpor or hypothermia [70]. A rectal probe attached to a type T thermocouple calibrated thermometer (Kent Scientific Corporation, Torrington, CT) with a resolution of 0.1˚C was used. Basal body temperature (BT) of mice housed in groups was recorded between 1530-1600h *i.e.*, before fasting and after 16hs of food deprivation *i.e.*, before allowing re-feeding.

Mice body composition (total body fat, lean mass, total body water) was determined in group-housed mice of both sexes at 10w, 20w and 30w of age by using the whole body quantitative magnetic resonance analyzer EchoMRI-500™ system (EchoMRI LLC, Houston TX), essentially as described [71].

## Statistics

Data obtained were plotted and analyzed by using Prism v5 (GraphPad Software, San Diego, CA). Result are represented as mean values ± SEM, with the number of individual points (*n*), the difference in means (*dm*) ± SEM or interactions *F(Dfn,DFd)* and their *p* values. Statistical significance for a *p* value <0.05 was obtained with one-way or two-way analyses of variance (ANOVA), as appropriate, followed by the Tukey-Kramer *post-hoc* test. Time series data expected to follow circadian rhythms were analyzed by applying the least squares method to fit a sine wave *i.e.*, Cosinor [72, 73] by using the publicly available web-based application *Cosinor. Online* (cosinor.online/app/cosinor.php), as defined in [74].

## Results

### Body weight accrual and body composition of female and male mice

The results of Fig 1 show that male mice accumulate significantly more BW over time than females, as expected. The increase in BW observed in males was significant at 5–6 weeks of age (males are 17.4% heavier than females, *n* = 10–18, *dm* = 4.25 ± 0.89g, *p*<0.01) and continued to increase throughout the first 20w of age to stabilize thereafter (CV: 0.10 ± 0.01). In fact, the BW of 30-35w old males remained ~20% higher than that of females (*dm* = 6.26 ± 1.05g, *p*<0.001). The absolute and relative contributions of lean and fat mass to total BW are shown in Fig 1B–1E. Absolute lean mass increased as a function of age in social males and females, but it was significantly higher in males at all ages tested (Fig 1B). Absolute fat mass, however, increased only in males over time and remained higher than that of females, reaching statistical difference at 20-30w of age (Fig 1C). When adjusted to total body mass, relative lean and fat mass were significantly reduced and increased, respectively, in 30w old males relative to females (Fig 1D and 1E, respectively). Therefore, fat mass contributes more than lean mass to total BW in the case of males housed in groups relative to that of females, as exemplified by the fat:lean ratio (Fig 1F).

### Energy intake of social mice

Energy intake was continuously monitored during 3 weeks in group-housed mice starting at 10w of age, *i.e.*, when the fat:lean ratio is similar in mice of both sexes (Fig 1F) and at 20-30w of age, when it increases only in males (Fig 1F). Mean net daily energy intake per mouse was not different between males and females at 10w or 30w of age [8.29 ± 0.18 and 7.86 ± 0.18 kCal/mouse or 9.30 ± 0.20 and 9.55 ± 0.16 kCal/mouse, *p* = 0.376 or *p* = 0.912, respectively] but significantly increased in 20w old males relative to females [9.52 ± 0.17 *vs*. 8.85 ± 0.17 kCal/mouse (*p*<0.05, *F*(14,183) = 11.99, *p*<0.001)] (S3A Fig). In line, net energy intake per mouse cumulated over 14 days was significantly increased in males, but only when mice were 20w old (S3C–S3E Fig). Because changes in BW and body composition alter long-term energy intake and covariate analysis has demonstrated that BW is a reliable independent variable to normalize energy intake [60, 75, 76], food consumption of mice of different ages and sexes were compared after adjusting net energy intake to total BW. The results shown in Fig 2A demonstrates that 20-30w old females consume significantly more food per unit of BW than age-matched males (*p*<0.05).

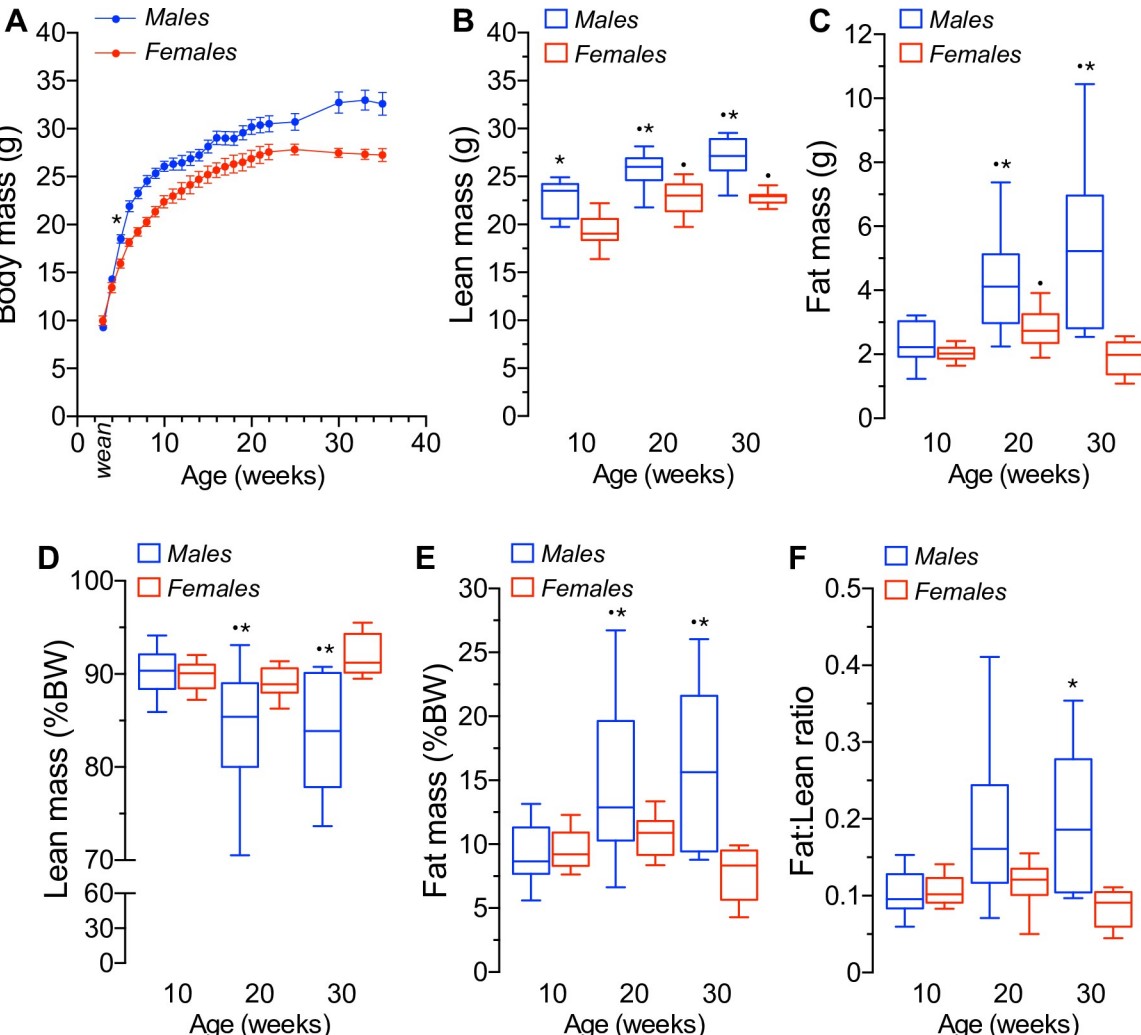

**Fig 1. Body mass accrual and body composition of normal mice housed in groups. A.** Weekly body mass in grams (g) of mice fed ad libitum a chow diet from weaning (post-natal day 19–20) to 35 weeks of age. Body mass is significantly higher in males than females from week 5 (n = 10, dm = 4.14 ± 0.89g, *p<0.001) and thereafter. **B.** Absolute lean mass of male and female mice at 10w, 20w and 30w of age (*p<0.01 sex; •p<0.001 10w age; F(2,84) = 0.79, p = 0.45). **C.** Absolute fat mass of 10w, 20w and 30w old male and female mice (*p<0.001 sex; •p<0.001 vs. 10w old, F(2,77) = 8.73, p<0.001). **D.** Lean mass of male and female mice at 10w, 20w and 30w of age relative to their mean body mass (*p<0.05 sex; •p<0.05 vs. 10w old, F(2,84) = 4.73, p<0.05). **E.** Fat mass relative to mean body mass of 10w, 20w and 30w old male and female mice (*p<0.05 sex; •p<0.05 vs. 10w old, F(4,137) = 9.25, p<0.001). **F.** Absolute fat mass (C) to absolute lean mass (B) ratio of male and female mice (*p<0.01 sex; •p<0.05 vs. 10w old mice, F(2,83) = 4.38, p<0.05).

The relative contribution of each photoperiod of the day to daily energy intake in social mice was next determined. As expected, Fig 2B shows that males and females consume ≥75% of their net daily energy intake during the scotophase. There were no sex-related diurnal or nocturnal differences in energy intake per unit of BW in 10-20w old mice (Fig 2B). However, 30w old female mice consumed significantly more food during both phases of the day than age-matched males (Fig 2B, *p*<0.05). There were no evident disruptions in the circadian pattern of energy intake between males and females (Fig 2C) beyond those expected in 30w old mice due to significant changes in nocturnal and/or diurnal energy intake *e.g.*, the mesor line of 30w old mice and the amplitude of 20w old males, as shown in Table 1. Similarly, no major changes in spontaneous ambulatory activity between males and females were observed (Fig

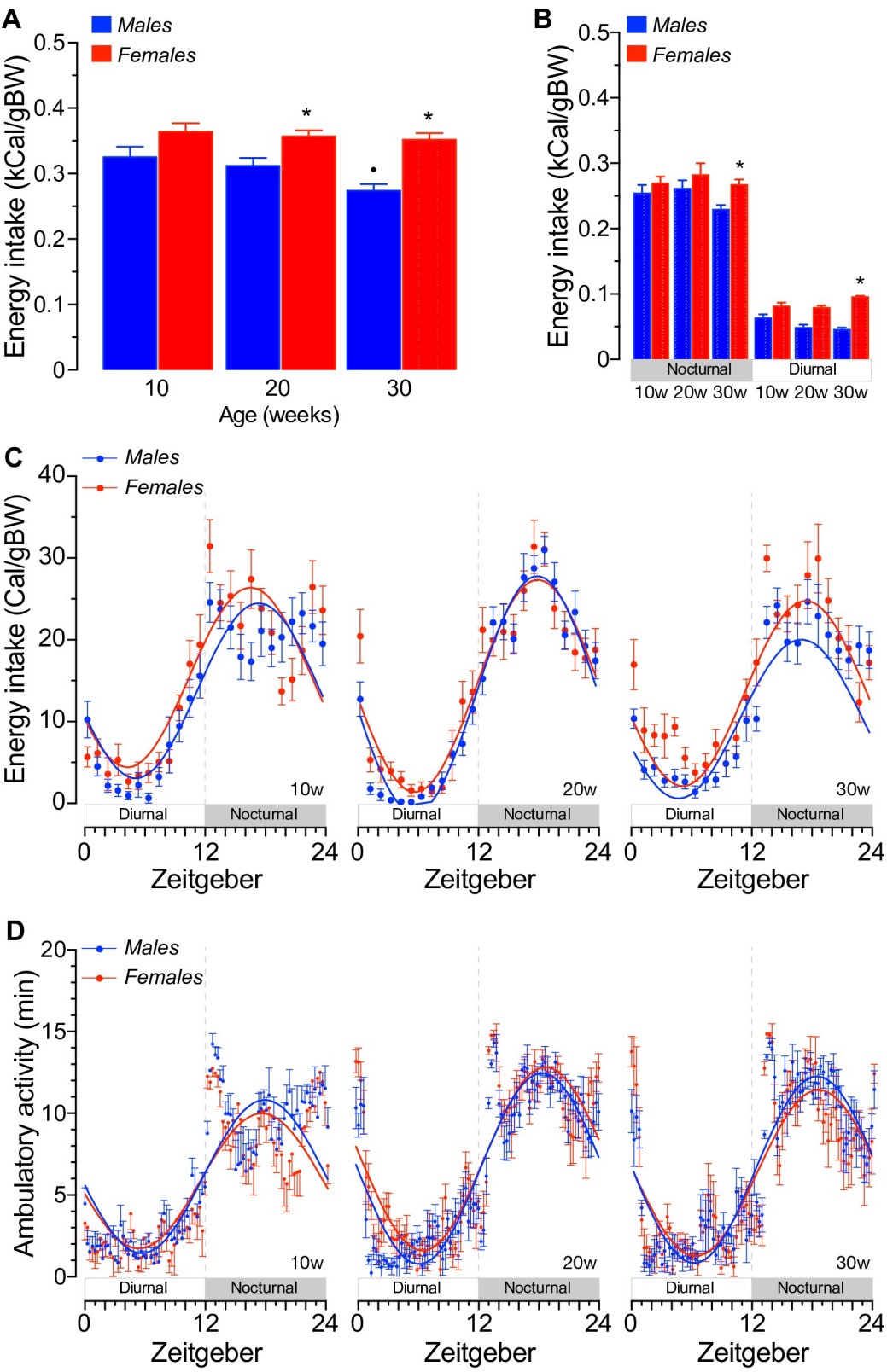

**Fig 2. Daily energy intake and ambulatory activity of normal mice housed in groups. A.** Daily energy intake of 10w, 20w and 30w old males and females. Intake was continuously computed during 14 days in mice fed ad libitum of a chow diet (3.0 kCal/g). Results are expressed as the mean ± SEM after adjusting to the mean BW recorded during the same

period of time (n = 10, *p<0.05 sex; •p<0.05 vs. 10w old mice, F(14,183) = 1.79, p<0.05). **B.** Mean adjusted nocturnal and diurnal energy intake of group-housed mice at 10w, 20w and 30w of age (n = 10, *p<0.05 sex, F(14,183) = 1.79, p<0.05). **C.** Circadian variation and cosinor analysis of adjusted energy intake in males and females of the indicated ages continuously computed in bins of 60min during 14 days of *ad libitum* feeding. Data for cosinor analysis is shown in Table 1. **D.** Circadian variation and cosinor analysis of net random ambulatory activity of mice of the indicated sexes and ages. Data represents the mean ± SEM of daily activity recorded throughout the course of 14 days of *ad libitum* feeding.

2D) beyond a significant increase in the activity of 20w old females relative to younger females (S3B Fig). Therefore, when taken together these results suggest that: *i*) 10-20w old social males show reduced nocturnal and diurnal energy intake per body mass when compared to females, although these differences were not statistically significant, *ii*) the combined nocturnal and diurnal energy intake of 10-20w old males results in a significant reduction in daily energy intake per unit of BW relative to age-matched females, but only at 20w of age, *iii*) daily energy intake adjusted to BW in 30w old social female is significantly higher than that of males due to significantly increased nocturnal and diurnal energy intake, *iv*) the circadian pattern of energy intake and spontaneous ambulatory activity is similar in social male and female mice at all ages tested.

### The body weight recovery and energy intake responses of fasted mice

Energy restriction imposed by prolonged fasting results in BW loss [77, 78] and hypothermia [79] due to increased net energy expenditure. Therefore, fasting of mice is expected to increase their energy intake upon re-feeding to recover the lost BW. Fig 3A–3C show that BW of 16hs fasted social males and females are significantly reduced relative to baseline (*p*<0.001), and to a similar extent, at all ages tested. However, the expected fasting-related hypothermia was age-dependent in females but not in males and significantly more pronounced in 10-20w old females than that of age-matched males (S4A Fig) uncovering potential sex- and age-related differences in thermogenesis/energy expenditure in mice housed in groups. In line, 10w and 30w old females recovered their lost BW within 2hs of re-feeding whereas males required ≥4hs to do so (Fig 3A and 3C). The magnitude of BW recovery while re-feeding was significantly higher in females than in males during the first 2-4hs at all ages tested (*p*<0.01), suggesting that females recover their lost BW more effectively than males. In fact, Fig 3B shows that 20w old females significantly exceeded their basal BW within 2-4hs of re-feeding (*p*<0.01), whereas group-housed males did not.

Fasted mice increase their hunger after fasting and their net energy intake upon re-feeding, even to a greater extent than that observed in the fed state [77, 80, 81]. However, the early differences in BW recovery were not related to sex- or age-dependent differences in the net energy intake of 16hs fasted mice during the first 8hs of re-feeding, as shown in Fig 3D–3F. In

**Table 1.**

| | Males | Females | Males | Females | Males | Females |
|---|---|---|---|---|---|---|
| **Age (w)** | 10 | | 20 | | 30 | |
| **Mesor (Cal/gBW)** | 13.40 ± 0.06 | 15.11 ± 0.06 | 13.42 ± 0.03 | 14.51 ± 0.06 | 11.54 ± 0.07 | 14.59 ± 0.04* |
| **Amplitude (Cal/gBW)** | 11.48 ± 0.08 | 11.23 ± 0.12 | 14.68 ± 0.03• | 13.98 ± 0.04 | 11.24 ± 0.08 | 11.47 ± 0.04 |
| **Acrophase (Zg)** | 16.6 ± 0.5 | 16.0 ± 0.2 | 17.3 ± 0.2 | 16.2 ± 0.4 | 16.0 ± 0.6 | 16.8 ± 0.2 |
| **Bathyphase (Zg)** | 4.6 ± 0.5 | 4.0 ± 0.2 | 5.3 ± 0.2 | 5.3 ± 0.4 | 4.0 ± 0.6 | 4.8 ± 0.2 |

*p<0.05 sex

•p<0.05 *vs*. 10w old.

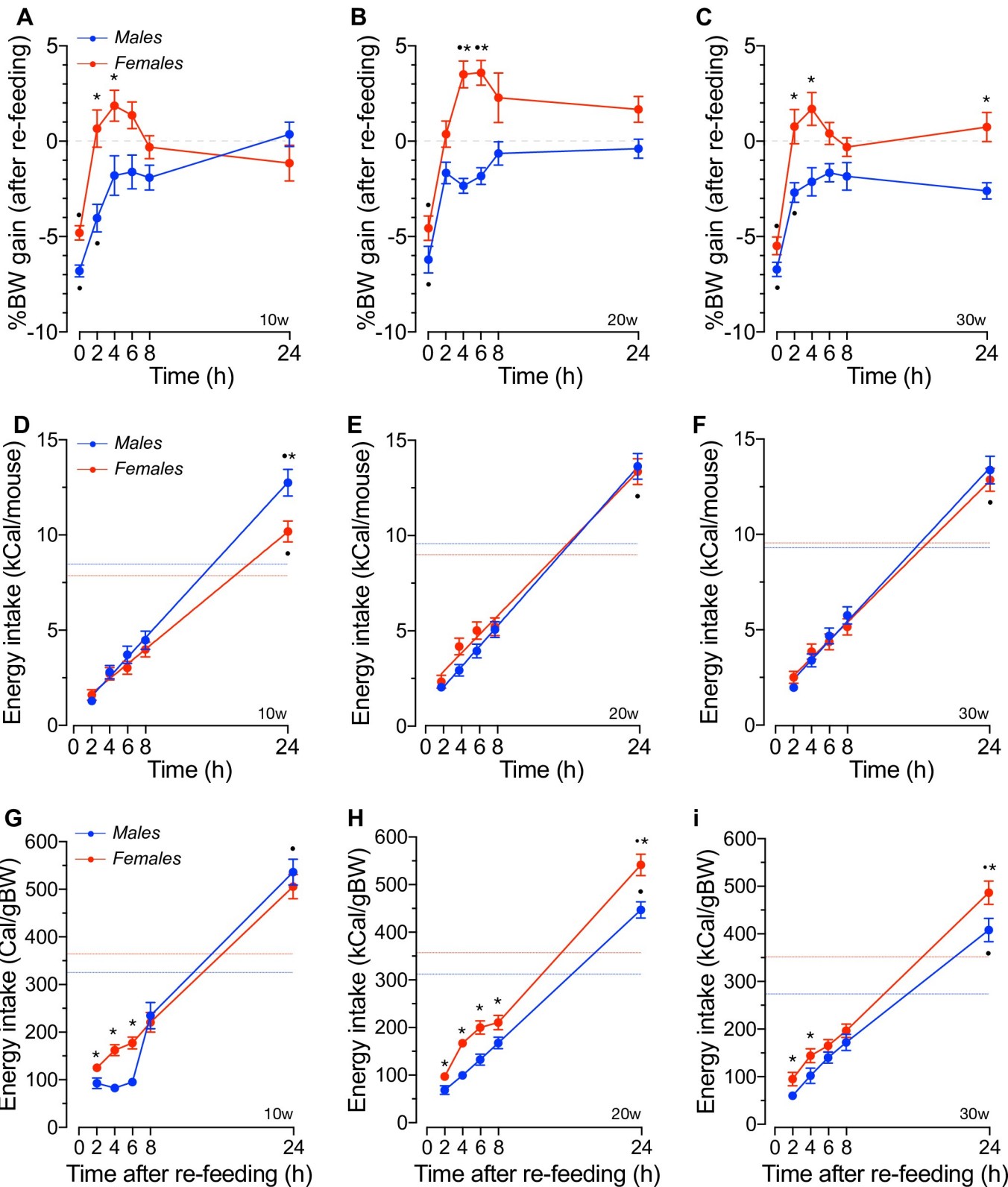

**Fig 3. Body mass and energy intake responses of group-housed mice to fasting. A-C.** Shown are BW accrual relative to baseline i.e., BW before fasting (dashed lines), recorded at the indicated times after allowing re-feeding of 16hs fasted males and female mice at 10w (A), 20w (B) and 30w (C) of age. Results are expressed as the mean ± SEM (n = 10, *p<0.05 sex; •p<0.01 vs. baseline, F(42,366) = 4.54, p<0.001). **D-F.** Net energy intake of 16hs fasted mice of both sexes at

10w (D), 20w (E) and 30w (F) of age recorded at the indicated time-points after allowing re-feeding. Results represent mean values ± SEM relative to the net daily energy intake obtained during the fed condition (baseline, dashed lines). **G-I.** Energy intake adjusted to BW at the indicated time-points after allowing re-feeding of 16hs fated mice of both sexes at 10w (G), 20w (H) and 30w (I) of age. Results are presented as mean values ± SEM. Dashed lines represents daily adjusted energy intake in the fed state [n = 10, $^*$p<0.01 sex; $^•$p<0.01 vs. baseline intake, $F_{(28,305)}$ = 3.02, p<0.001].

fact, net energy intake significantly increased after 8hs of re-feeding to reach and/or exceed *ad libitum* values ($p < 0.05$). Further, 10w old males ate significantly more than females (Fig 3D, $p < 0.05$). These results resemble previous data obtained in single-housed ~12w old males and females [77] or 22-24w old males [80, 81] of the C57BL/6J background housed under standard *vivarium* conditions. When energy intake was adjusted to BW at each time point during re-feeding, the results shown in Fig 3G–3I demonstrate that social males eat consistently and significantly less calories per unit of BW than females during the first ~2-6hs of re-feeding ($p < 0.05$).

The difference in the rate of BW recovery and that of net or normalized energy intake of mice during the first hours of re-feeding prompted us to calculate their feed efficiency (FE) *i.e.*, the change in BW per unit change in energy intake (ΔgBW/ΔkCal) and metabolic efficiency (ME) *i.e.*, the change in energy intake needed to change 1g in BW (ΔgkCal/ΔgBW). The results, shown in Fig 4, demonstrate no significant sex- or age-related differences in FE at all re-feeding time points tested. This was also the case for ME during the first 8hs of re-feeding (Fig 4D–4F). However, ME was significantly increased in 10-20w old females 24hs after re-feeding whereas that of 30w old females was reduced relative to males ($p < 0.05$). Of note, the results in Fig 4D–4F also demonstrate that the ME of 30w old males is significantly increased relative to younger males ($p < 0.05$) whereas that of females remained unchanged at all ages tested. Therefore, collectively these results suggest that: *i*) social male and female mice decrease their BW to a similar extent in response to 16hs fasting, *ii*) 10w old females recover BW twice as faster than age-matched males independently of their net energy intake, *iii*) 20-30w old mice recover their lost BW to baseline levels at a similar rate but re-fed females increase their basal BW, *iv*) fasting increases net 24hs energy intake per mouse without changing FE at all ages tested, and *v*) 10-20w old females are metabolically more efficient than males, but ME significantly increases in older males.

## The definition of a single meal in group-housed mice

To gain insights into the behavioral control of energy intake in group-housed mice, we first determined the intermeal interval (IMI) threshold (IMI$t$). To that end, we followed similar criteria originally validated by Zorrilla *et al.* in single-housed rats [31] and mice [30]. For instance, the rate of change in nocturnal meal size was calculated as a function of assumed IMI at 30s increments by subtracting the number of calories of a meal obtained at a given IMI to that of the previous 30s [30, 31] in 20w old male mice housed in groups ($n = 10$) for 10 days. Fig 5B shows that the rate of change in meal size reaches a minimum when IMI = 300s, which is identical or very close to those previously demonstrated in single-housed mice [30] and rats [31]. To confirm these results, nocturnal meal frequency and meal size were computed as a function of IMI in undisturbed 10, 20 and 30w old male and female mice housed in groups. The results shown in Fig 5C–5F demonstrate that IMI$t$ ≥5min defines the minimum IMI at which meal frequency (Fig 5C and 5D) and meal size (Fig 5E and 5F) rates of change decrease or increase, respectively, as a function of IMI. Therefore, an IMI$t$ ≥5min is here used to define meals and the feeding microstructure of 10-30w old mice of both sexes, housed in groups and fed a chow diet.

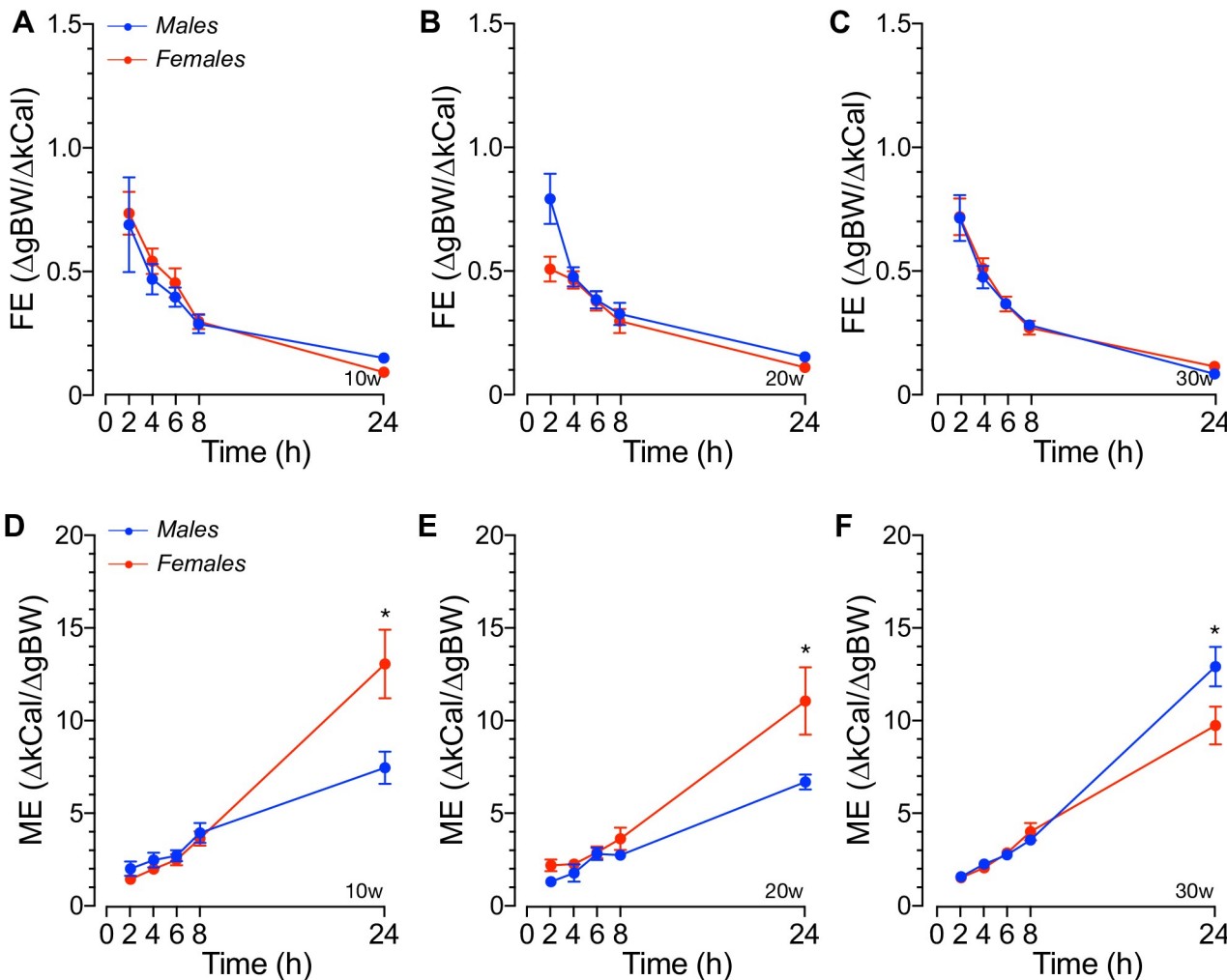

**Fig 4. Feed and metabolic efficiency of group-housed mice in responses to fasting. A-C.** Feed efficiency (FE), calculated as the ratio between the changes in body weight (ΔBW) and the net energy intake (ΔkCal) recorded at the indicated times after allowing re-feeding in 16hs fasted male and female mice at 10w (A), 20w (B) and 30w (C) of age. **D-F.** Metabolic efficiency (ME) calculated as the inverse of FE at the indicated times after allowing re-feeding of 16hs fasted male and female mice of 10w (D), 20w (E) and 30w (F) of age. Results represent the mean ± SEM (n = 10, *p<0.05 males vs. females).

## Meal size and frequency in social male and female mice fed ad libitum

We first tested the hypothesis that the increased daily caloric intake per unit of BW observed in 20-30w old female mice relative to males (Fig 2A) is related to sex-dependent changes in meal size, meal frequency or both. The results shown in Fig 6A demonstrate that 10w old female mice ate significantly larger meals than males but only during the scotophase, whereas 20w old females ate significantly larger meals than males during both photoperiods of the day (*p*<0.01). Notably, no age-dependent changes in nocturnal meal size of males and females were noted. However, meal size of 10-20w old females and 20w old males (*p*<0.05) were significantly reduced during the light phase. There was no statistical difference in meal size between 30w old males and females, either during the dark or during the light phase.

Based on the previous results and to account for the daily energy intake of male and female mice (Fig 2A and 2B), we next tested the hypothesis that meal frequency compensates, at least in part, for the sex- and age-related differences in meal size. The results shown in Fig 6B reveal

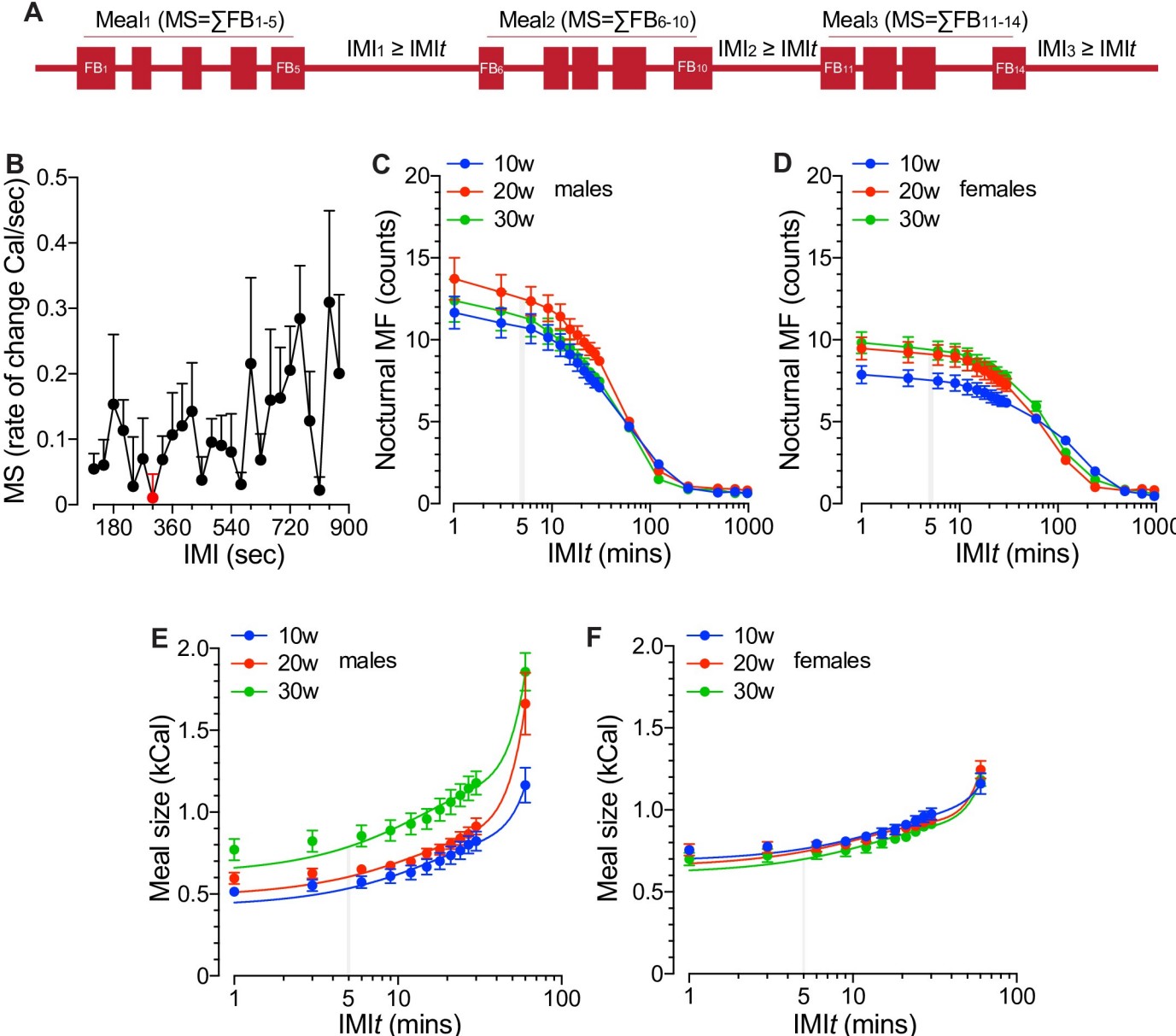

**Fig 5. The feeding microstructure of group-housed mice and its definitions. A.** Schematization of the feeding microstructure of mice. Represented are feeding bouts (FBs, filled squares) clustered in meals separated from each other by a non-eating intermeal interval (IMI) of equal or longer duration than a minimum IMI threshold (IMIt). **B.** Determination of the minimum rate of meal size (MS) change (Cal/sec) as a function of increasing intermeal intervals (IMI, sec) in 20w old male mice housed in groups (n = 10) and fed ad libitum undisturbed for 7 consecutive days. Data are expressed as the mean ± SEM. The minimum value is indicated as a red dot (IMIt = 300sec). **C-F.** Validation of the IMIt to indicate changes in meal frequency (C-D) and meal size (E-F) in male (C, E) and female (D, F) mice fed undisturbed a chow diet for 7 days. Shown are data corresponding to the nocturnal photoperiod of the day. The number of meals and meal size were determined as a function of increasing IMI (min) to visualize if the IMIt (5min) distinguishes changes in their rate of change. As shown, IMIt ≥5mins serve as a threshold to mathematically define a meal in 10w, 20w and 30w old male and female mice housed in groups.

that 10-20w old females consume significantly less number of meals than males ($p<0.01$), but only during the night (Fig 6B, *left panel*). Diurnal meal frequency was in fact reduced relative to the scotophase, as expected, but not different between males and females (Fig 6B, *right panel*). In addition, there was no significant age-related differences between the number of nocturnal and diurnal meals, beyond a tendency to increase during the night in 30w old

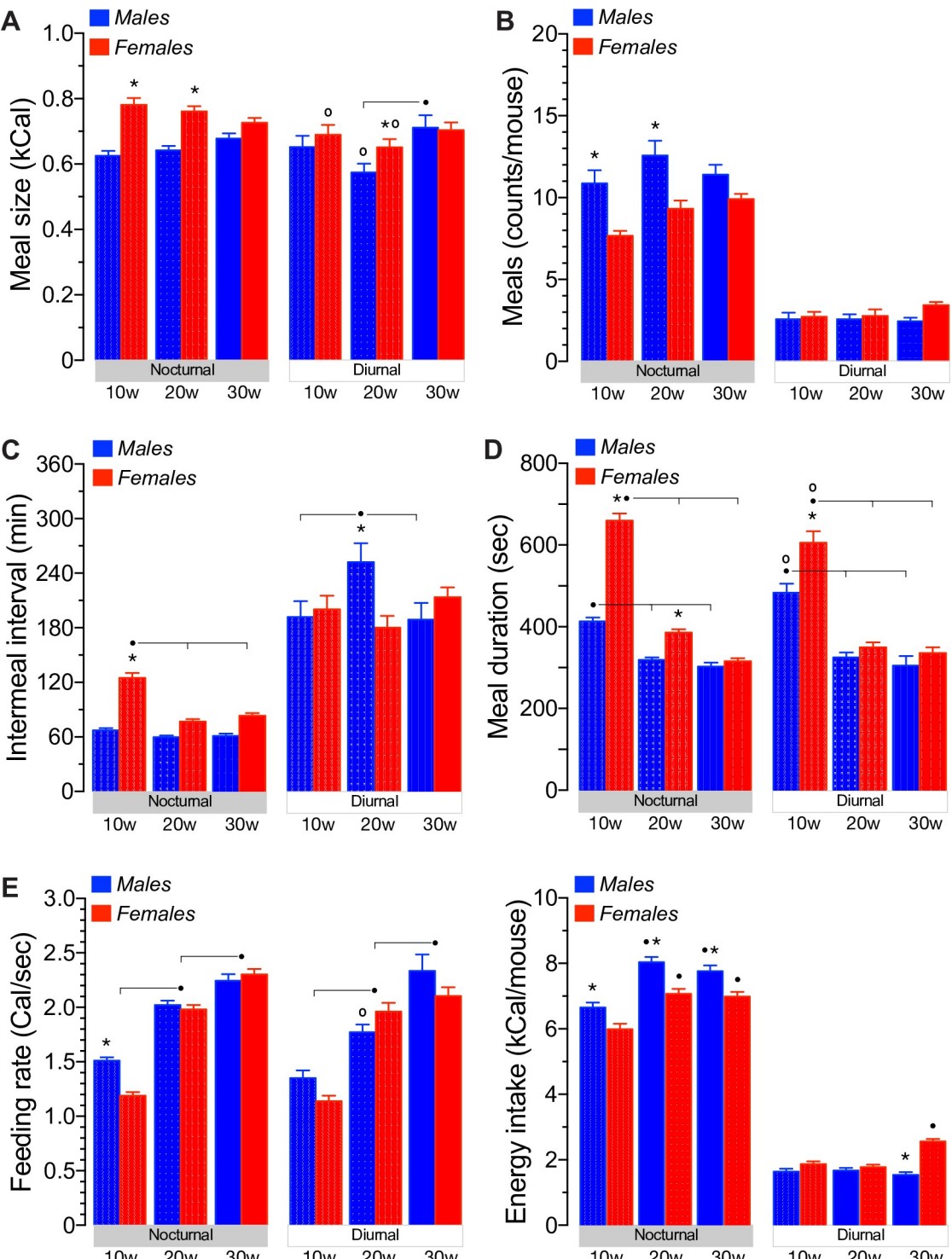

**Fig 6. The feeding microstructure of social mice feed ad libitum. A-F.** Shown are the meal size [A, kCal, (n = 10, *p<0.05 sex; •p<0.05 age; °p<0.05 phase, F(14,183) = 2.01, p<0.05)], number of meals [B, counts per mouse, (n = 10, *p<0.05 sex, F(14,183) = 7.16, p<0.001)], intermeal interval [C, minutes, (n = 10, *p<0.05 sex; •p<0.05 age, F(14,183) = 5.81, p<0.001)], meal duration [D, seconds, (n = 10, *p<0.05 sex; •p<0.05 age; °p<0.05 phase, F(14,183) = 2.60, p<0.01)], feeding rate [E, calories per second, (n = 10, *p<0.05 sex; •p<0.05 age; °p<0.05 phase, F(14,183) = 1.13, p = 0.33)] and the net energy intake calculated as the product of mean meal size and frequency [F, kCal, (n = 10, *p<0.05 sex; •p<0.05 age, F(14,183) = 11.99, p<0.001)] of group-housed male and female mice at 10w, 20w and 30w of age fed ad libitum a chow diet. Data was recorded during 14 consecutive days and separated according to the two photoperiods of the day.

females. These results, when taken together, suggest that: *i*) during the scotophase, meal frequency compensates, at least in part for the differences in meal size observed in males and females, particularly in 10-20w old mice, and *ii*) the significant decrease in diurnal meal size observed in 20w old males is not compensated by increased diurnal meal frequency or *vice versa* for females. Therefore, nocturnal and diurnal meal size and frequency differentially contribute to the total daily energy intake of social male and female mice over time. That is, the comparable daily energy intake of 10week old social male and female mice relates to their smaller and larger meals consumed at a higher and lower frequency, respectively, during the nocturnal and diurnal phases of the day. However, the increased daily energy intake of 20w old female mice relative to males relates to significantly larger nocturnal and diurnal meals.

## The intermeal interval of social male and female mice fed ad libitum

Shorter intermeal intervals, without significant changes in meal size is expected to increase meal frequency and energy intake. Therefore, we tested the hypothesis that differences in daily energy intake between social male and female mice are also related to changes in the non-eating time spent between meals. As shown in Fig 6C (*left panel*), the IMI recorded during the scotophase is significantly longer in 10w old females relative to age-matched males ($p<0.001$). Of note, there were no age-dependent changes in the nocturnal IMI of males or in the diurnal IMI of females (Fig 6C). However, the nocturnal IMI of females significantly shortened ($p<0.01$) at 20-30w of age to reach levels indistinguishable from those of age-matched males (Fig 6C, *left panel*). As expected for a reduced number of meals, social males and females spent significantly longer time between meals during the daylight than during the night. However, contrary to females, the diurnal IMI in social males was significantly lengthened in 20w old mice relative to younger and older males ($p<0.01$), and relative to females at all ages tested ($p<0.01$). Therefore, when taken together these results suggest that: *i*) 10w old social females spend longer time between meals than males, but only during the night, *ii*) nocturnal IMI shortens in older females but not in males, and *iii*) diurnal IMI significantly increases in 20w old males but remains unchanged in females at all ages tested.

## The meal size and intermeal interval re-feeding responses of fasted mice

Fasted male and female mice allowed to re-feed after 16hs of food deprivation did not increase the size or latency of their first meals or their duration (S5 Fig), resulting in no significant changes in the rate of feeding when compared to the fed *ad libitum* condition. Nevertheless, Fig 7A shows that 16hs fasting increases the average meal size significantly above *ad libitum* levels in 10w old females ($p<0.05$), but only during the first 2-4hs of re-feeding. Meal size of 10-20w old mice did not change relative to baseline (Fig 7B and 7C), but that of 20w old females was significantly increased relative to age-matched males ($p<0.01$) only during the first 4hs of re-feeding (Fig 7B). Further, meal duration did not differ from *ad libitum* baseline or between sexes in social mice at all ages tested (Fig 7D–7F). When meal size and meal duration were related, the feeding rate showed minimum changes relative to baseline or between sexes in all mice, as shown in Fig 7G–7I. Collectively these results suggest that prolonged fasting does not trigger significant sex- or age-dependent changes in meal size in social mice.

Therefore, we next tested the hypothesis that the non-eating time spent between meals determines re-feeding energy intake in fasted social mice. As shown in Fig 8A–8C, the number of meals consumed in response to fasting linearly increased as a function of the re-feeding time in male and female mice at all ages tested. Although the increase in meal frequency rate were sex- and age-independent, the net number of meals accrued by males during 24hs of re-feeding was significantly increased relative to *ad libitum* ($p<0.01$) whereas that of females was

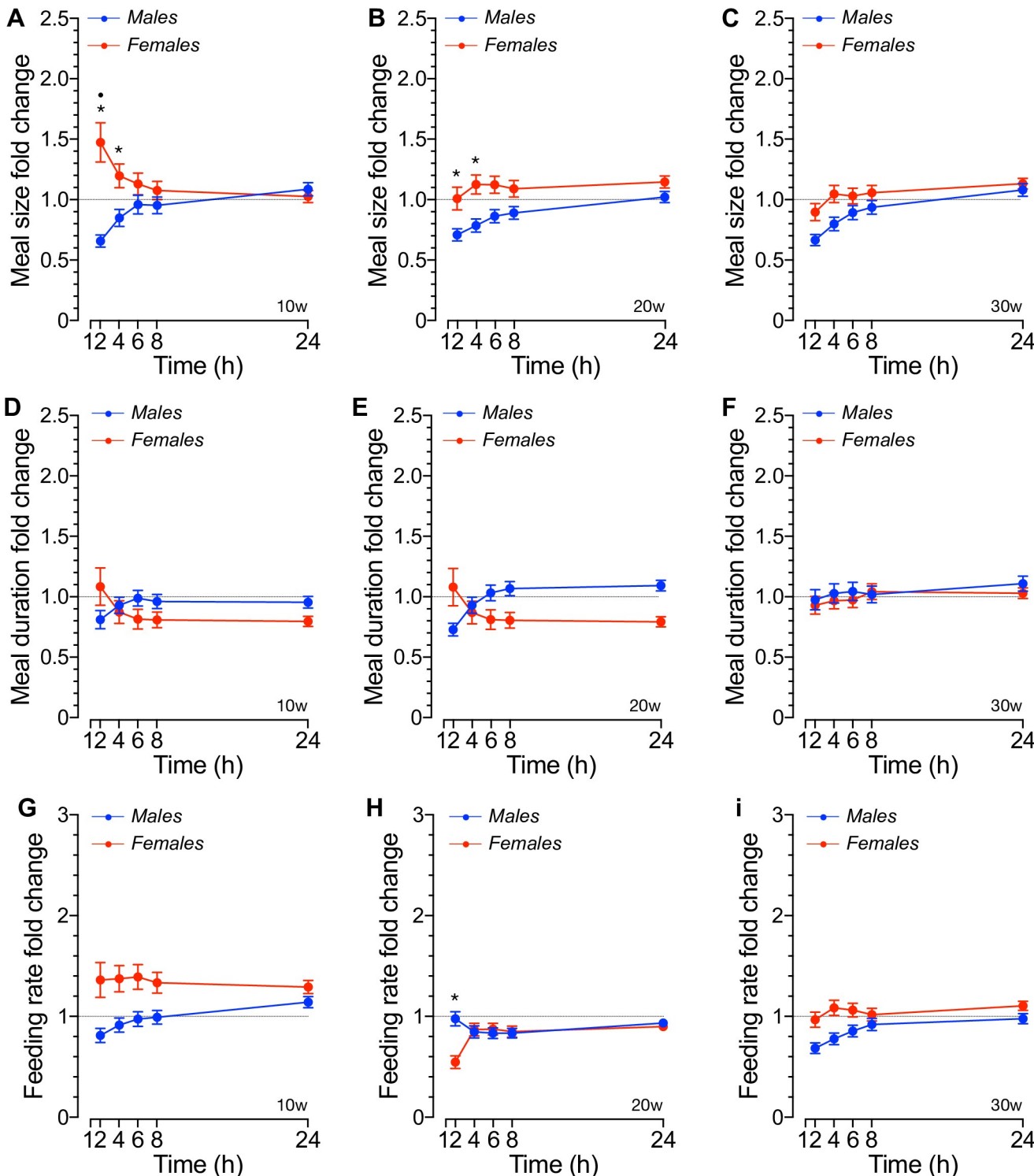

**Fig 7. Meal size, meal duration and feeding rate changes in responses to fasting and re-feeding in social mice. A-C.** Meal size (kCal) recorded at the indicated times after allowing re-feeding of 16hs fasted male and female mice at 10w (A), 20w (B) and 30w (C) of age [n = 10, *p<0.05 sex; •p<0.05 vs. baseline, F(28,305) = 1.64, p<0.05]. **D-F.** Meal duration or mean time spent in each single meal recorded at the indicated times after allowing re-feeding of 16hs fasted mice of both sexes at 10w (D), 20w (E) and 30w (F) weeks of age. **G-I.** Mean feeding rate calculated as the ratio between meal size (A-C) and meal duration (D-F) [n = 10, *p<0.05 sex, F(28,305) = 1.01, p = 0.46]. The results are expressed as mean values ± SEM relative to baseline (dashed lines) i.e., the mean daily meal size, duration or rate during the fed state.

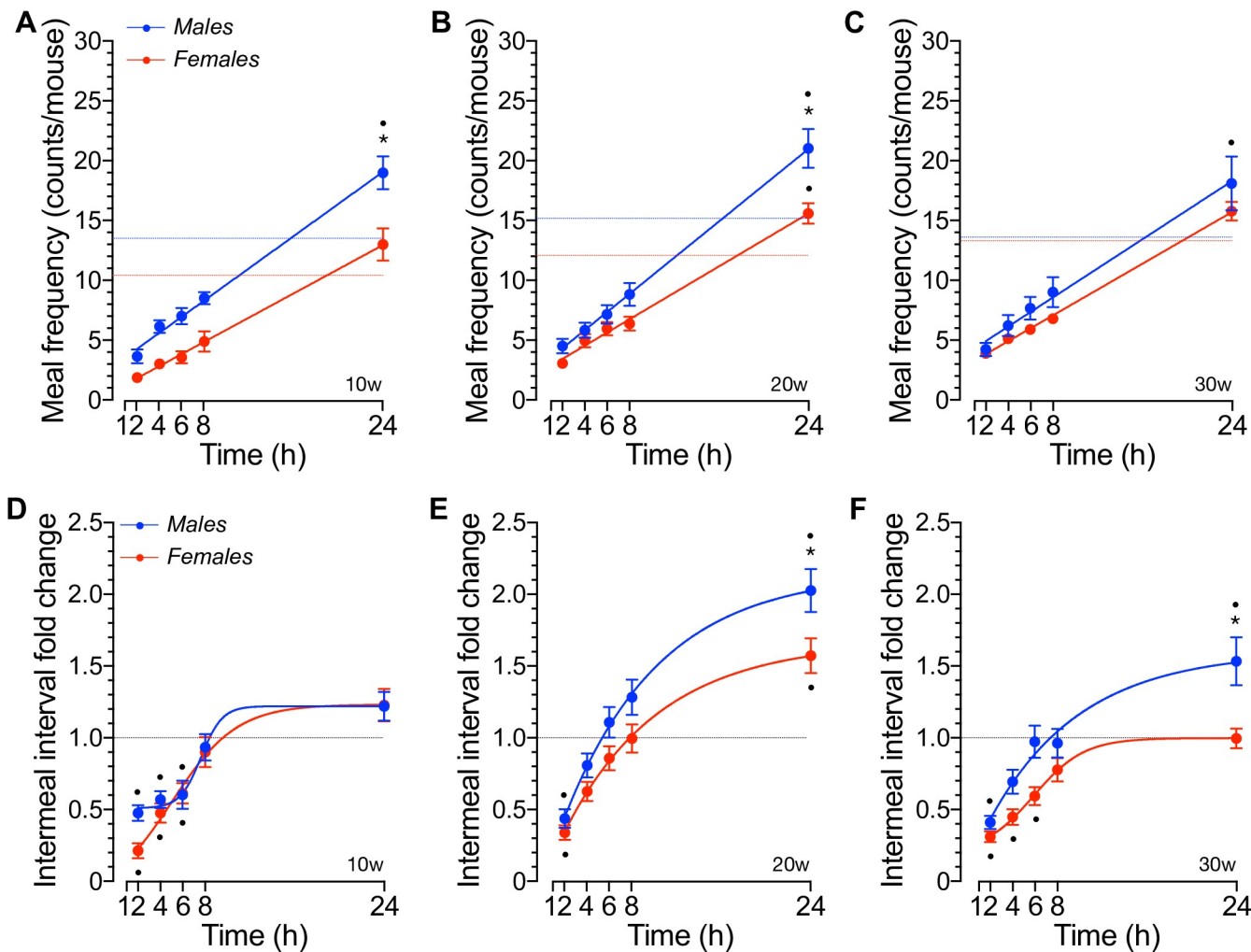

**Fig 8. Meal frequency and intermeal interval changes in responses to fasting and re-feeding in social mice. A-C.** Meal frequency (counts/mouse) recorded at the indicated time-points during re-feeding in 16hs fasted male and female mice of 10w (A), 20w (B) and 30w (C) of age [n = 10, *p<0.05 sex; •p<0.05 vs. baseline, F(35,366) = 2.57, p<0.001]. **D-F.** The mean intermeal interval (min) calculated as the time difference between the moment a meal is initiated and the time at which the previous meal has ended was recorded at the indicated time-points during re-feeding of 16hs fasted male and female mice of the indicated ages [10w (A), 20w (B) and 30w (C), n = 10, *p<0.05 sex; •p<0.05 vs. baseline, F(28,305) = 1.329, p = 0.128]. Results represent mean values ± SEM relative to baseline (dashed lines) i.e., mean daily number of meals per mouse and IMI during the fed state.

only significant at 20w of age (Fig 8A–8C, $p$ = 0.040). Therefore, these results suggest that fasting reduces the non-eating time between meals in social mice. To verify and extend that suggestion, we next determined the IMI responses of mice to fasting and re-feeding. Fig 8D–8F show that, relative to the fed IMI, the non-eating time spent between meals is robustly and significantly shortened to similar extents in males and females during the first 2hs of re-feeding ($p$<0.01). In fact, the IMI remained significantly shortened during the first ~6hs of re-feeding relative to baseline in 10w old mice, to gradually reach fed levels thereafter (Fig 8D). However, the reduction in IMI of 20w old mice lasted significantly less time thus reaching baseline values faster than those of 10w old mice (Fig 8E). Further, the IMI of 20w old mice significantly exceeded baseline levels 24hs after re-feeding (Fig 8F, $p$<0.05) and the 24hs IMI of 20w and 30w old male mice were significantly lengthened relative to that of females (Fig 8E and 8F, $p$<0.01). Therefore, these results suggest that: *i*) food deprivation significantly and robustly shortens the non-eating time spent between meals in both male and female mice, *ii*) the

shortening of the time spent not eating lasts significantly longer in 10w old mice that in older mice, *iii*) the re-feeding response of mice to fasting gradually prolongs the IMI to reach or exceed baseline in an age-dependent manner.

## Discussion

Although there is little information regarding the role of age and sex in BW gain and energy regulation in group- *vs.* single-housed mice, our results suggest that the relative increase in BW accrual of social males (Fig 1A) is the combined result of increased accumulation of lean and fat mass, as one may have expected. In fact, our experiments demonstrated that fat mass accrual has a more prominent and variable age-related effect in the net BW increase of males than of females, as previously shown [54, 60]. The mean age-dependent increase in BW of social males and females and their mean lean and fat mass accrual (Fig 1B and 1C) were also similar to that of single-housed C57BL/6J male and female mice [60, 76] or pair-housed male mice [53] fed normal chow diets. These effects were apparent already at 20w of age (Fig 1B and 1C). Therefore, these results suggest that housing male or female mice of the C57BL/6J background in groups plays a minimal role in their net BW accrual or body composition relative to single-house mice and that fat mass accrual of social males is significantly higher and variable than those of females. These interpretations are in line with previous suggestions indicating that the variation in BW/composition is sex-dependent and potentially linked to differential energy conversion to BW [59, 60]. Indeed, group-housed male and female mice differentially increased their net and cumulative daily energy intake in an age-dependent manner (S3 Fig), even to similar extents to those observed in single-housed mice [60]. Therefore, to better compare potential sex- and age-dependent changes in energy intake in grouped-housed mice, food intake data was adjusted to BW (Fig 2A). Our results demonstrated that daily energy intake per unit of BW was not different between 10w old males and females, but that of males significantly decreased over time (Fig 2A). This reduction was not related to obvious abnormalities or shifts in the circadian pattern of energy intake (Fig 2C and Table 1). In addition, the spontaneous ambulatory activity of mice housed in groups was not different between sexes (Fig 2D) and remarkably similar to that of group-housed C57BL/6J mice [82, 83]. Of note, 20w old group-housed females significantly increased their net nocturnal and diurnal ambulatory activity when compared to younger females (S3B Fig), which correlated with increased adjusted daily energy intake (Fig 2A). Therefore, the age/sex-dependent changes in adjusted energy intake cannot be directly attributed to sex- or age-related differences in the circadian pattern of food intake or activity in mice housed in groups. Nevertheless, subtle sex- and/or age-related effects on energy intake and/or its circadian pattern cannot be simply discarded when the sample size is relatively small [84].

At this point, it is worth noting that locomotor activity is considered to play a minor role in metabolic rate/energy expenditure in single- or group-housed 20w old C57BL/6J mice [85]. However, a recent large-scale study found a strong correlation between energy expenditure and locomotor activity [76]. In connection with that and within the limitations of our experiments imposed by the comparatively small number of male and female mice ($n = 10$) used, our results suggest that acutely eliminating $\geq 70\%$ of mice' net daily energy intake resulted in ~5–7% BW loss (Fig 3A–3C), as noted in previous studies using single-housed C57BL/6J mice [77]. Although these data suggest that a similar caloric restriction reduces metabolic rate or net energy expenditure to a similar extent in social males and females, fasting-induced hypothermia was significantly more pronounced in 10-20w old females (S4A Fig), as previously reported for single-caged mice [86, 87]. Since fat mass contributes to whole-body metabolism by regulating thermogenic activity [88] and males housed in groups showed increased fat

mass, our results suggest that the thermoregulatory mechanisms involved in heat production/ loss is sex- and age-dependent in house grouped mice. Further, these potential thermoregulatory differences appeared not related to changes in the ambulatory activity of mice while fasting (S4C Fig). Therefore, our results support the hypothesis that 10w-20w old males may suppress heat loss during fasting to a greater extent than females, implying that group-housed males may reduce their net energy expenditure to a lesser extent than that of females during fasting, as suggested in single-housed mice [89, 90]. Of note, in the fed state, single-housed C57BL/6J female mice [91–93] and females of most mammalian species [94] show higher BT than males, a difference that became larger in older mice due to an age-dependent decrease in males' BT (S4B Fig). Although the latter effect was not observed in single-housed males [92], altogether, these data support the notion that sex differentially contributes to the thermoregulatory mechanisms of mice housed in groups.

Even though the increase in energy intake observed in older female mice relative to that of males may be explained, at least in part, based on the thermodynamics of energy homeostasis, an extra layer of complexity is noted when sex- and age-related changes in meal size, frequency or intermeal intervals are observed [31, 46]. Our results demonstrated that IMI$t \geq$5min defines a meal in social mice of both sexes (Fig 5) offering for the first time an adequate framework to compare our results with previous data. Indeed, Tabarin *et al*. used an IMI$t \geq$6min to describe the short-term feeding microstructure of ~30w old single-caged male mice of mixed genetic background [30]. We also used IMI$t \geq$6min and IMI$t \geq$8min to determine the feeding microstructure of social mice, but the results were not significantly different from those obtained by using IMI$t \geq$5min (S2 Fig). Although we have used fewer animals to determine IMI$t$ relative to that of previous work ($n$ = 24, [30]), it is important to note that by imposing an IMI$t \geq$5-6min we and others [26, 30, 31] could visualize the behavioral satiety sequence including grooming, wandering, playing, exploration, sniffing, resting and other non-feeding activities lending strong physiological support to the criteria used to define meals and determine the feeding behavior of social mice.

During the night, when mice consume most of their daily meals (Fig 6B), 10w old females had a meal every ~2hs whereas age-matched males spent ~1h to eat a new meal (Fig 6C). In addition, these females spent significantly longer time eating bigger meals than 10w old males (Fig 6D). Some, or all, of these sex-related differences in the feeding behavior of grouped-housed mice were lost in 20w or 30w old mice, respectively. Indeed, nocturnal IMI and feeding rate were not different between 20w old males and females and no sex- or phase-related differences in the feeding microstructure of 30w old were recorded. Few significant changes in diurnal feeding behavior were noted: meal duration remained significantly higher in 10w old females (Fig 6D) whereas meal size and IMI were significantly increased and reduced, respectively, in 20w old females (Fig 6A and 6C). When these results are compared to those obtained in single-housed mice using a similar IMI$t$ criteria, it becomes apparent that housing 20w old mice in groups reduces their nocturnal and diurnal meal size by ~20% [*i.e*., nocturnal: 0.642 ± 0.013 kCal (Fig 6A) *vs*. 0.518 ± 0.038 kCal and diurnal: 0.575 ± 0.026 kCal (Fig 6A) *vs*. 0.467 ± 0.032 kCal [30]] but not the diurnal-related decrease in meal size, which was ~9% in both studies. Although the mean meal size of housed-grouped females has not been determined previously, some comparative conclusions may be drawn from a previous study using IMI$t \geq$5min [26]. For instance, the nocturnal liquid meal size of single-housed ~10w old females was ~25% bigger than that of males, an estimation close to that of social mice at similar ages (~20%, Fig 6A left panel). Therefore, female mice appear to consume more calories per meal than males, which in our experiments became readily apparent at 20w of age (Fig 6A). However, the nocturnal IMI of individually housed 10w old mice was not different between males and females when fed a liquid diet, and shorter [26] than those of group-housed mice

(Fig 6C). Further, the nocturnal IMI was also shorter in individually housed ~20w old males [30]. Therefore, mice housed in groups spend longer time not eating than individually-housed mice during the night. Consistent with the fact that very few meals are consumed during the inactive phase of the day (Fig 6B), the diurnal IMI was significantly prolonged in social mice (Fig 6D), which was also noted in individually-housed 10w old mice fed a liquid diet [26]. In fact, the increase in IMI was exaggerated in 20w old males; the time spent not eating was significantly longer than that of females (Fig 6C, right panel). In summary, sex, age and housing influence the feeding behavior of mice under stable energy balance when fed *ad libitum* a chow diet. These differences are more apparent under negative energy balance. Indeed, our results demonstrated that the hyperphagic response to a negative energy balance imposed by prolonged fasting does not delay or inhibit satiation beyond a transient increase in the re-fed meal size of 10w old females (Fig 7A), an effect that disappeared in older females (Fig 7B and 7C). Instead, fasted mice housed in groups control fasting-induced hyperphagia by increasing the number of meals to a similar extent in males and females (Fig 8A–8C) and significantly shortening the time between meals (Fig 8D–8F). However, these responses were qualitatively and quantitatively sex- and age-dependent. Indeed, the IMI lengthened as the re-feeding time increased following a sigmoidal kinetics that saturated at or above basal fed levels in 10w (Fig 8D) or in 20-30w old males (Fig 8E and 8F), respectively, thus indicating that re-feeding increases satiety in social males in an age-dependent. However, such responses were significantly reduced (Fig 8E) or lost (Fig 8F) in 20w and 30w old females, respectively. Therefore, these results suggest that social mice control fasting-induced hyperphagia by engaging age- and sex-dependent satiety mechanisms rather than those involved in satiation, as one might have expected [95, 96].

## Summary

Many behavioral, physiological, neural and humoral signals are involved in the long- and short-term control of energy intake [27, 29, 95, 96]. An extra layer of complexity is now added by our findings suggesting sex-, age-, and cage-dependent differences in the behavioral control of energy intake. The differences in the size of single meals (satiation) or the time spent not eating (satiety) between group- and single-housed mice, or even among sexes and ages may be related to the experimental design, the relatively small sample size [84], the genetic background of mice (mixed *vs.* C57BL/6J) [97], liquid *vs.* solid food, or housing [55, 98, 99]. In this latter respect, ~10w old C57BL/6J mice showed enhanced anxiety-like behavior when individually housed [55, 100] whereas stress/anxiety did reduce meal size in mice of that genetic background and age [101]. Although the physiological mechanisms involved in the control of satiation and satiety in mice remain poorly defined [29], it has been well-documented that estrogens inhibit energy intake by reducing meal size [46, 102–104] and that adiposity signals including insulin and leptin may control meal size [25], even in a sex-dependent manner [105–107]. However, recent data has not provided evidence of a modulatory role of estrogens on meal size [108] and a potential long-term modulatory relationship between adiposity signals and satiation (meal size) and/or satiety (IMI) awaits demonstration. The use of validated criteria to study the feeding pattern of social mice offers for the first time a conceptual blueprint and methodological framework to test those hypotheses and elucidate the mechanisms involved in the control of satiation and satiety in mice.

## Supporting information

**S1 Fig. Experimental design timeline.** The study design is shown as a 2wk interval timeline spanning the entire experiment (~35 weeks). As indicated, ten male and female mice were

weaned at week 3 of age (p19-20). When mice were not monitored for their feeding behavior, they were housed in groups of 5 mice per cage per sex. When mice were 8-9wk old, they were implanted with unique RFID chips. Blue arrowheads indicate the time where male or female mice housed in groups of 5 animals were randomly swapped between their 2 respective cages. This was done a week before and immediately after monitoring their feeding behavior, which are represented as red rectangles. Before and during the 3 weeks of monitoring their feeding behavior, male or female mice were housed in groups of 10, as indicated by green arrowheads. Purple arrowheads point out the times where body composition was determined.
(PDF)

**S2 Fig. The diurnal feeding microstructure of group-housed mice computed by using IMI ≥5min and IMI ≥8min.** Shown are the meal size (A, kCal), frequency (B, counts per mouse), duration (C, seconds) and intermeal interval (D, minutes) of group-housed male and female mice at 10w, 20w and 30w of age fed ad libitum a chow diet. Diurnal data was recorded during 14 consecutive days by using two IMI threshold criteria: ≥5min and ≥8min. Results are expressed as the mean ± SEM.
(PDF)

**S3 Fig. Net and cumulated daily energy intake and ambulatory activity of normal mice housed in groups. A.** Net energy intake (kCal/mouse) of 10w, 20w and 30w old males and females continuously recorded during 14 days while fed *ad libitum* a chow diet (3.0 kCal/g). Results represent the mean ± SEM ($n = 10$, $^*p<0.05$ sex; •$p<0.05$ *vs.* 10w old mice). **B.** Mean nocturnal and diurnal ambulatory activity integrated from data shown in Fig 2D. Results are expressed as the mean ± SEM ($n = 10$, •$p<0.05$ *vs.* 10w old mice). **C-E.** Energy intake of 10w (C), 20w (D) and 30w (E) old males and females cumulated over 14 days of *ad libitum* feeding a chow diet. Dashed lines indicate the mean net daily energy intake ($^*p<0.05$ sex).
(PDF)

**S4 Fig. Fasting-induced hypothermia and cumulated ambulatory activity during fasting in normal mice housed in groups. A.** Shown are the net core body temperature (BT) change in group-housed male and female mice in response to 16hs of fasting (A) and basal BT obtained in mice fed *ad libitum* (C). Results represent the mean ± SEM ($n = 10$, $^*p<0.001$ sex; •$p<0.05$ *vs.* 10w old mice). **C.** Ambulatory activity of 10w, 20w and 30w old male and female during 16hs of fasting.
(PDF)

**S5 Fig. First meal size and duration of mice fed ad libitum or after re-feeding. A-B.** Shown are the mean first nocturnal meal size (A, kCal) and duration (B, min) of group-housed male (n = 10) and female (n = 10) mice at 10w, 20w and 30w of age fed a chow diet ad libitum (blue bars) or after re-feeding following a fasting period of 16hs (red bars). Results represent the mean ± SEM.
(PDF)

## Acknowledgments

Part of the results presented here constitutes YR Master of Science Thesis [2020, MS, Wright State University, Pharmacology and Toxicology]. We are grateful to Drs. J. Ashot Kozak and Khalid Elased (WSU) for their valuable comments during the development of this project.

## Author Contributions

**Conceptualization:** Mauricio Di Fulvio.

**Data curation:** Mauricio Di Fulvio.

**Formal analysis:** Yakshkumar Dilipbhai Rathod, Mauricio Di Fulvio.

**Funding acquisition:** Mauricio Di Fulvio.

**Investigation:** Yakshkumar Dilipbhai Rathod, Mauricio Di Fulvio.

**Methodology:** Yakshkumar Dilipbhai Rathod, Mauricio Di Fulvio.

**Project administration:** Mauricio Di Fulvio.

**Resources:** Mauricio Di Fulvio.

**Supervision:** Mauricio Di Fulvio.

**Validation:** Yakshkumar Dilipbhai Rathod, Mauricio Di Fulvio.

**Visualization:** Yakshkumar Dilipbhai Rathod, Mauricio Di Fulvio.

**Writing – original draft:** Mauricio Di Fulvio.

**Writing – review & editing:** Yakshkumar Dilipbhai Rathod, Mauricio Di Fulvio.

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
