## [Decision Letter · Decision Letter 0]

26 Nov 2020

PONE-D-20-31618

The feeding microstructure of male and female mice

PLOS ONE

Dear Dr. Di Fulvio,

First, I apologize for taking a significant amount of time to evaluate your manuscript.  Because of the pandemic, I have had difficulty securing the reviewers who are willing to review in a timely manner. I would like to thank the two reviewers who did extensive reviews of this manuscript. Although the reviewer #1 suggested to accept this manuscript with minor revision, reviewer #2 recommend to reject the manuscript for publication in PLoS ONE. From my understanding, a primary reason for the reviewer #2 is that the study is a descriptive, but doesn’t provide mechanisms for satiety or satiation. But the reviewer #2 also indicated that the methods used to measure feeding patterns are technically sound, which is in agreement with reviewer #1 that the study is likely to provide a comprehensive reference for other studies and provide a validated framework in group-housed mice for measuring IMI. A descriptive study is within the scope of PLoS ONE, if experiments and analysis are performed to high technical standards (PLOS ONE: accelerating the publication of peer-reviewed science). Therefore, I invite you to submit a revised manuscript. Both reviewers indicated a number of suggestions. Please address all of the issues raised by the two expert reviewers. The reviewer expressed concern about the small sample size in the current study. As the reviewer suggested, please acknowledge this limitation in the discussion.

I also have several suggestions, those are mostly editorial.

Please adhere to the ARRIVE guidelines.

The ARRIVE guidelines 2.0.

https://journals.plos.org/plosbiology/article?id=10.1371/journal.pbio.3000410

Please provide an approximate number of inter breeding generations of the mice you used in this study. Provide detailed information (e.g. day light intensity of apparatus).

Line 159, please remove “(not shown)”. I suggest to show the data in Supporting Information, but I think simply removing “(not shown)” is fine. Please note: PLoS ONE doesn’t allow statements which are supported by unpublished data.

Line 434 and Line 547 (not shown), please indicate analysis in the Supporting Information.

Please remove grant information from the acknowledgment. Grants typed in Financial Disclosure will appear in the front page as Funding.

It looks like energy intake and ambulatory activity in 10W old mice are bimodal. Cosinor analysis may not be an appropriate analysis to determine the phase and amplitude of the rhythm. There is no need to address this concern. I just want to let you know this.

We look forward to receiving your revised manuscript.

Kind regards,

Shin Yamazaki, Ph.D.

Section Editor

PLOS ONE

Journal Requirements:

2. Please include a separate caption for each figure in your manuscript.

3. Please ensure that you refer to Figures 1-8 in your text as, if accepted, production will need this reference to link the reader to the figure.

Reviewers' comments:

Reviewer's Responses to Questions

**Comments to the Author**

1. Is the manuscript technically sound, and do the data support the conclusions?

Reviewer #1: Yes

Reviewer #2: Partly

2. Has the statistical analysis been performed appropriately and rigorously? 

Reviewer #1: Yes

Reviewer #2: No

3. Have the authors made all data underlying the findings in their manuscript fully available?

Reviewer #1: Yes

Reviewer #2: Yes

4. Is the manuscript presented in an intelligible fashion and written in standard English?

Reviewer #1: Yes

Reviewer #2: Yes

5. Review Comments to the Author

Reviewer #1: The goal of this study was to investigate sex- and age-related differences in the feeding patterns of group-housed C57BL/6J mice. The authors used an automated feed intake monitoring system to measure meal patterning, including meal size and intermeal interval, which are markers of satiation and satiety. The authors carefully and comprehensively characterize and report BW, body composition, and meal patterning in male and female mice at 10wks, 20wks, and 30wks during ad libitum feeding and fasting-refeeding. These data are likely to provide a comprehensive reference for other studies and provide a validated framework in group-housed mice for measuring IMI. The data and discussion also nicely outline age- and sex-dependent differences and similarities between single-housed (prior studies) and group-housed (current study) mice. The main concern is the small sample size. Only 4 cages of mice were studied, and possible influences of social structure (e.g., fighting/dominant-submissive mice) on individual mice were not considered/discussed.

Main Concerns/suggestions:

1. According to the methods, a total of 20 mice were studied: 10 males and 10 females. The mice were housed in groups of 5, so that 4 cages of mice were studied in total. There is concern that this is a small sample size, with only 2 replicates for each sex of the social group. This could be discussed as a limitation.

2. It would be interesting to separately plot data from individual mice that are housed in the same cage together. The authors describe that mice were randomly re-grouped before beginning the experiment. This is likely to cause fighting in the males, perhaps as intended by the authors to simulate real social dynamics. The male mice were also highly variable in their lean and fat mass. Was this variability within cages/social groups (i.e., greater body weight in dominant male)? Or was one social group smaller that the other cage? Was low variability in BW and lean/fat mass in females due to low/absent fighting?

3. Throughout the manuscript, it would be more accurate to use “sex” instead of “gender” to describe the mice since gender is a term reserved for humans. Gender is a social construct while sex refers to biological and physiological sex.

Introduction

1. Line 31-33: It would be helpful to include more references for the statement about alterations in feeding patterns regulating adiposity and BW in animals and humans. Only 1 reference for a human study is included. Alternatively, “animals” could be removed from the sentence.

2. Lines 37-39: Reference #10, Sherman et al does not support the authors’ statement, because the magnitude of calories consumed was not similar between ad lib and RF mice.

Results/Figures:

1. Fig. 2: The mesor (dotted horizontal lines) are shown only for only one age, 30 wks. The figure legend indicates that the mesor is shown for all data. Alternatively, the mesor could be removed from the figure since it is reported in Table 1.

2. The meal patterning data are described (Fig. 6) in the context of the total food intake (fig 2A) and diurnal vs. nocturnal food intake (Fig 2B). It may be helpful to combine these panels into one figure, rather than flipping back and forth between Fig 2 and Fig 6 to consider that data together.

3. It would also be helpful to group the fasting-refeeding food intake data (Fig. 3) with the fasting-refeeding meal patterning data (Fig. 7), perhaps by making them successive figures (e.g. Figs 6 and 7).

Discussion:

1. Line 511: There may be an error in this sentence, should it say “similar to that of single-housed C57 mice”?

2. The discussion is thorough and nicely lays out similarities and differences between prior studies of single-housed and/or liquid-fed mice and the current study of group-housed, pellet-fed mice.

Line 316: typo, redeeding should be refeeding

Line 444: typo, test should be tested

Line 536: in older mice due and age-dependent decreased, error in this sentence

Reviewer #2: The present manuscript aimed at describing the microstructure of feeding patterns in mice housed in a social environment of 5 mice / cage. A comparison is performed between males and females and at three different ages; 10, 20 and 30 weeks of age. In order to monitor feeding a sophisticated method was used using individual radio-frequency transponders that permitted to follow individual activity of collectively housed mice, combined with a Feed and Water intake monitoring system that allows to identify the food intake with a resolution of 0.001g. With this system authors measured meal frequency, meal duration, meal size (kCal and Cal/min), inter meal intervals (IMI), energy intake, body weight increase and body weight composition. Main processes of satiety and satiation were defined with this procedure. Moreover, a response to a 16h food deprivation was assessed.

The methods used for reporting feeding patterns are technically sound and provide a series of graphs and results that provide clear description of the feeding patterns of grouped-house mice at three different ages.

A main concern about this study is that in the last paragraph of the introduction authors offer to provide “a better insight of the contribution of satiation (meal size) and satiety (IMI) for the control of food intake…” This is however not analyzed nor further discussed. Along their manuscript and especially in the summary (or conclusions?) authors compare their findings with previous studies reporting feeding patterns in single housed mice. However in this paper and under their conditions this is not confirmed because the single housed mice were not run in this study.

In the methods it is indicated that one week before starting experiments mice were regrouped to allow “high-order social hierarchies and behaviors”. This is not further analyzed or discussed. The summary (or conclusion) provides a discussion about a series of variables including hormonal signals, mechanisms for satiation, adiposity,that may influence the mechanisms of feeding patterns, but were not assessed.

Importantly the values obtained with the ANOVA are not provided.

Brief: This study provides a description of feeding patterns according to 3 ages and sex in group-housed mice, which does not differ from single housed mice (not included in the experimental design). The study does not provide further mechanisms and does not relate their measurements with possible mechanisms for satiety or satiation.

6. PLOS authors have the option to publish the peer review history of their article (what does this mean?). If published, this will include your full peer review and any attached files.

Reviewer #1: No

Reviewer #2: No

---

## [Author Response · Author response to Decision Letter 0]

16 Dec 2020

Responses to Critiques

Responses to Editor's comments:

"Please adhere to the ARRIVE guidelines."

"Please provide an approximate number of inter breeding generations of the mice you used in this study. Provide detailed information (e.g. day light intensity of apparatus)."

Response: We have provided the information requested and carefully followed the guidelines for reporting animal research to include all the details needed to replicate our findings.

"Line 159, please remove “(not shown)”. I suggest to show the data in Supporting Information, but I think simply removing “(not shown)” is fine. Please note: PLoS ONE doesn’t allow statements which are supported by unpublished data."

"Line 434 and Line 547 (not shown), please indicate analysis in the Supporting Information."

Response: As advised, we have removed "(not shown)" in lines 169, 434 and 547 and provided the pertinent information in new Supplementary Figures 1 and 4 

"Please remove grant information from the acknowledgment. Grants typed in Financial Disclosure will appear in the front page as Funding."

Response: Grant information in the Acknowledgment section has been removed

"It looks like energy intake and ambulatory activity in 10W old mice are bimodal. Cosinor analysis may not be an appropriate analysis to determine the phase and amplitude of the rhythm. There is no need to address this concern. I just want to let you know this."

Response: We do appreciate this valuable comment. 

Responses to comments of Reviewer #1

"The main concern is the small sample size. Only 4 cages of mice were studied, and possible influences of social structure (e.g., fighting/dominant-submissive mice) on individual mice were not considered/discussed."

"1. According to the methods, a total of 20 mice were studied: 10 males and 10 females. The mice were housed in groups of 5, so that 4 cages of mice were studied in total. There is concern that this is a small sample size, with only 2 replicates for each sex of the social group. This could be discussed as a limitation."

Response: We do apologize for the confusion. The experimental design was as follows: a total of 10 male and 10 female mice were kept in standard cages (5 mice/cage of 0.01m3) from weaning to ~8-9w of age. After that time, mice were transponded (RFID) and scrambled between the two standard cages and kept for an additional week in those two cages. Then, the total number of mice (10 males or 10 females) were placed in the apparatus' cage (HM2 System, 0.1m3) to test their feeding behavior for 3 weeks. After the end of that testing period, mice were randomly placed back into 2 standard cages (5 mice/cage 0.01m3) for additional ~6weeks. When mice reached ~19w of age, they were randomly scrambled again between the two cages and kept for an additional week in their room. After that time, ~20w old mice were placed in the apparatus' cage (n=10) to test their feeding behavior for 3 weeks. After the end of that second testing period, mice were randomly placed back into their two standard cages (5 mice/cage) for additional ~6weeks. When mice reached the age of ~30w, they were randomly scrambled for the third time and kept for an additional week in their room. Then, all mice were placed one last time into the apparatus' cage to test their feeding behavior for 3 weeks. Therefore, a total of 10 males and 10 females were evaluated for their feeding behavior at ~10w, ~20w and ~30w of age. In summary, when mice were not being tested for their feeding behavior, they were kept in 4 cages containing 5 mice each in their assigned room of the vivarium. We have now better explained the experimental design and noted their limitations in the discussion, as suggested.

"2. It would be interesting to separately plot data from individual mice that are housed in the same cage together. The authors describe that mice were randomly re-grouped before beginning the experiment. This is likely to cause fighting in the males, perhaps as intended by the authors to simulate real social dynamics. The male mice were also highly variable in their lean and fat mass. Was this variability within cages/social groups (i.e., greater body weight in dominant male)? Or was one social group smaller that the other cage? Was low variability in BW and lean/fat mass in females due to low/absent fighting?"

Response: The Reviewer's questions are very interesting. At this point, we do not have a precise answer for them. Mice were scrambled early in their life and were kept together in groups of 5 from weaning until behavioral experiments were performed, time at which 10 mice were grouped and tested. This setting was carried out until mice reached ~35w of age i.e., during the whole duration of the experiments. We have observed few fighting events in the standard cages (5 male mice/cage 0.01m3) when males were ~10w old, and even less events when these mice were in the cage of the HM2 apparatus (10 males/cage, 0.1m3). Yet, fighting episodes became rare as males aged. These observations are in line with previous results showing that fighting events in males decrease when the size of the cage increases [1], although this subject remains in debate [2]. Of note, we did observe increased feeding frequency in one of the males, which was the one who attained the highest body weight. Although these observations may be in line with previous data demonstrating increased feeding frequency in dominant male mice [1], we could not clearly ascertain the identity of the α male under our experimental conditions due to the low frequency of fights. In addition, the mouse with the highest body weight and feeding frequency was not the one eating first after fasting. Therefore, it remains possible that the increased body weight of that mouse is the result of increased feeding frequency rather than behavioral dominance. We have also recorded variability in the net caloric intake in all males, which could help explain, at least in part, the variability in male fat mass observed under our experimental conditions. 

"3. Throughout the manuscript, it would be more accurate to use “sex” instead of “gender” to describe the mice since gender is a term reserved for humans. Gender is a social construct while sex refers to biological and physiological sex."

Response: Following the Reviewer's advice, we now use the word "sex" instead of "gender". 

Introduction

"1. Line 31-33: It would be helpful to include more references for the statement about alterations in feeding patterns regulating adiposity and BW in animals and humans. Only 1 reference for a human study is included. Alternatively, “animals” could be removed from the sentence."

Response: Following the Reviewer's suggestion, we have added additional references to support the hypothesis that feeding patterns may participate in the regulation of adiposity and BW in animal models [2] and in humans [3].

"2. Lines 37-39: Reference #10, Sherman et al does not support the authors’ statement, because the magnitude of calories consumed was not similar between ad lib and RF mice."

Response: Indeed. We have excluded reference #10 from that statement.

Results/Figures:

"1. Fig. 2: The mesor (dotted horizontal lines) are shown only for only one age, 30 wks. The figure legend indicates that the mesor is shown for all data. Alternatively, the mesor could be removed from the figure since it is reported in Table 1."

Response: We have removed the mesor line in Fig 2. 

"2. The meal patterning data are described (Fig. 6) in the context of the total food intake (fig 2A) and diurnal vs. nocturnal food intake (Fig 2B). It may be helpful to combine these panels into one figure, rather than flipping back and forth between Fig 2 and Fig 6 to consider that data together."

Response: Following the Reviewer's suggestion, we have included an additional panel in Fig 6 (new Fig 6F) reflecting the product of meal size and meal frequency i.e., net energy intake during the nocturnal and diurnal periods of the day to improve data analysis. 

"3. It would also be helpful to group the fasting-refeeding food intake data (Fig. 3) with the fasting-refeeding meal patterning data (Fig. 7), perhaps by making them successive figures (e.g. Figs 6 and 7)."

Response: The rationale behind the apparently separated analysis was twofold. First, we wanted to find out sex/age-related differences in BW recovery after 16h fasting and its relationship to net/adjusted energy intake (Figs 3-4), independently of any potential contributions of satiation (meal size) and/or satiety (IMI) to energy intake under these conditions. Second, we wanted to relate changes in energy intake in re-fed mice (which were minor) to changes in the feeding microstructure of fasted mice. To achieve the second goal, we had to thoroughly assess and determine the feeding microstructure of mice under ad libitum conditions first (Figs 5-6), to have a reliable baseline from which to compare. There, it is the reason whereby Figs 5-6 precedes Fig 7. 

Discussion:

"1. Line 511: There may be an error in this sentence, should it say “similar to that of single-housed C57 mice”?"

Response: The error in the sentence was corrected.

"2. The discussion is thorough and nicely lays out similarities and differences between prior studies of single-housed and/or liquid-fed mice and the current study of group-housed, pellet-fed mice."

Response: We are grateful and appreciate this comment.

"Line 316: typo, redeeding should be refeeding"

"Line 444: typo, test should be tested"

"Line 536: in older mice due and age-dependent decreased, error in this sentence"

Response: All these errors were corrected.

Responses to Reviewer #2 comments

A main concern about this study is that in the last paragraph of the introduction authors offer to provide “a better insight of the contribution of satiation (meal size) and satiety (IMI) for the control of food intake…” This is however not analyzed nor further discussed.

Response: The objective of our work was to provide a reliable procedure to measure two key behavioral variables involved in the control of feeding in group-housed mice: satiation (meal size) and satiety (intermeal interval, IMI) following physiological and behavioral definitions of meal size already validated in single-caged mice of ~30w of age [3]. We believe that our experimental procedures can accurately determine the size of a single meal and the interval of time elapsed between them. We do acknowledge that these determinations do not elucidate the mechanistic/physiological/regulatory aspects of satiation and satiety. However, the accurate and reproducible quantification of meal size and IMI provide a framework on which new hypotheses related to the mechanisms of satiation and satiety can be tested in the future. Until now, that was not possible. 

"Along their manuscript and especially in the summary (or conclusions?) authors compare their findings with previous studies reporting feeding patterns in single housed mice. However in this paper and under their conditions this is not confirmed because the single housed mice were not run in this study."

Response: As mentioned, the objective of our manuscript was to provide a new blueprint/framework on which satiation and satiety could be accurately measured in house-grouped mice. In the discussion, we compare our results with those of single-house mice in order to better grasp the notion that the criteria used to define a meal in mice still stand and remain adequate to measure meal size and intermeal interval in group-housed mice of both sexes and at different ages. 

"In the methods it is indicated that one week before starting experiments mice were regrouped to allow “high-order social hierarchies and behaviors”. This is not further analyzed or discussed."

Response: It is well-recognized that mice housed in groups develop complex social dynamics and behavioral hierarchies [4, 5]. A thorough analysis of these complex behaviors has been already published for mice of the CD1 [6] and of the C57BL6/J [7] genetic backgrounds. We have added these additional references to direct the reader to these comprehensive studies and briefly discussed the subject in relation to fighting episodes observed under our experimental conditions. 

"The summary (or conclusion) provides a discussion about a series of variables including hormonal signals, mechanisms for satiation, adiposity,that may influence the mechanisms of feeding patterns, but were not assessed."

Response: As noted previously, we do acknowledge the fact that the mechanistic/physiological aspects of satiation and satiety were not the objectives of our work. Nevertheless, the recorded differences in meal size of grouped mice of both sexes, for instance, do point out to sex-dependent mechanisms involved in satiation, a hypothesis that could be tested at the mechanistic level in the future. In addition, and as another example, our results also suggest that prolonged fasting does not trigger significant sex- or age-dependent changes in meal size thus indicating that the satiation mechanisms triggered upon net caloric deficit are similar in male and female mice. Although a direct role of leptin (adiposity signal) in the control of meal size has been recently proposed in single-housed male rats [4], the contribution of this adiposity signal in the control of the feeding behavior in mice remains unknown.

"Importantly the values obtained with the ANOVA are not provided."

Response: Following this important point, we have now provided these values. 

"Brief: This study provides a description of feeding patterns according to 3 ages and sex in group-housed mice, which does not differ from single housed mice (not included in the experimental design)."

Response: Indeed, our study is the first description of the feeding pattern of mice housed in groups and provide evidence in support of the hypothesis that housing may have a minor impact in the feeding behavior of mice. Yet, that hypothesis does relate to meal size (satiation) of 20-30w old male mice (when a comparative analysis between our results and those of others is considered). Although we do not know whether these results also apply to female mice or mice of different genetic backgrounds or ages or diets, our results do suggest that mice housed in groups spend longer time not eating (reduced satiety) than individually-housed mice during the night. Therefore, sex, age and housing influence the feeding behavior of mice under stable energy balance when fed ad libitum a chow diet. 

"The study does not provide further mechanisms and does not relate their measurements with possible mechanisms for satiety or satiation."

Response: As recently suggested: "...exploratory research should also be encouraged within the framework of open science." [8]. We do share that opinion, which was the driver to submit our results to Plos One. Descriptive experimentation has value, in particular when exploratory research has the immediate potential for discoveries. In our specific case, we believe that our results stimulate the generation of new hypotheses aimed at determining the physiology (or mechanisms) of satiation and satiety, at any level. These new hypotheses can now be tested, as we are providing, and making available to others, the blueprint to adequately and reproducibly measure satiation and satiety in the most common animal model housed in their social environment.

References

1. Lee W, Yang E, Curley JP. Foraging dynamics are associated with social status and context in mouse social hierarchies. PeerJ. 2018;6:e5617. https://doi.org/10.7717/peerj.5617. PMID: 30258716

2. Espinosa-Carrasco J, Burokas A, Fructuoso M, Erb I, Martin-Garcia E, et al. Time- course and dynamics of obesity-related behavioral changes induced by energy-dense foods in mice. Addict Biol. 2018;23(2):531-43. https://doi.org/10.1111/adb.12595. PMID: 29318700

3. Jenkins DJ, Wolever TM, Vuksan V, Brighenti F, Cunnane SC, et al. Nibbling versus gorging: metabolic advantages of increased meal frequency. N Engl J Med. 1989;321(14):929-34. https://doi.org/10.1056/NEJM198910053211403. PMID: 2674713

4. So N, Franks B, Lim S, Curley JP. A Social Network Approach Reveals Associations between Mouse Social Dominance and Brain Gene Expression. PLoS One. 2015;10(7):e0134509. https://doi.org/10.1371/journal.pone.0134509. PMID: 26226265

5. Peters SM, Pothuizen HH, Spruijt BM. Ethological concepts enhance the translational value of animal models. Eur J Pharmacol. 2015;759:42-50. https://doi.org/10.1016/j.ejphar.2015.03.043. PMID: 25823814

6. Williamson CM, Franks B, Curley JP. Mouse Social Network Dynamics and Community Structure are Associated with Plasticity-Related Brain Gene Expression. Front Behav Neurosci. 2016;10:152. https://doi.org/10.3389/fnbeh.2016.00152. PMID: 27540359

7. Theil JH, Ahloy-Dallaire J, Weber EM, Gaskill BN, Pritchett-Corning KR, et al. The epidemiology of fighting in group-housed laboratory mice. Sci Rep. 2020;10(1):16649. https://doi.org/10.1038/s41598-020-73620-0. PMID: 33024186

8. Thompson WH, Wright J, Bissett PG. Open exploration. Elife. 2020;9. https://doi.org/10.7554/eLife.52157. PMID: 31916934

---

## [Decision Letter · Decision Letter 1]

19 Jan 2021

PONE-D-20-31618R1

The feeding microstructure of male and female mice

PLOS ONE

Dear Dr. Di Fulvio,

I asked two original reviewers to review your revised manuscript. Although reviewer #2 suggested to accept current version of manuscript for publication in PLoS ONE, reviewer #1 indicated several minor issues. Please address those issues.

We look forward to receiving your revised manuscript.

Kind regards,

Shin Yamazaki, Ph.D.

Section Editor

PLOS ONE

Reviewers' comments:

Reviewer's Responses to Questions

**Comments to the Author**

1. If the authors have adequately addressed your comments raised in a previous round of review and you feel that this manuscript is now acceptable for publication, you may indicate that here to bypass the “Comments to the Author” section, enter your conflict of interest statement in the “Confidential to Editor” section, and submit your "Accept" recommendation.

Reviewer #1: (No Response)

Reviewer #2: All comments have been addressed

2. Is the manuscript technically sound, and do the data support the conclusions?

Reviewer #1: Yes

Reviewer #2: Yes

3. Has the statistical analysis been performed appropriately and rigorously? 

Reviewer #1: Yes

Reviewer #2: Yes

4. Have the authors made all data underlying the findings in their manuscript fully available?

Reviewer #1: Yes

Reviewer #2: Yes

5. Is the manuscript presented in an intelligible fashion and written in standard English?

Reviewer #1: Yes

Reviewer #2: Yes

6. Review Comments to the Author

Reviewer #1: The reviewers have addressed most of my concerns. I still have several remaining suggestions to address my concerns.

1. Abstract: The abstract describes the feeding microstructure results in a very general and vague way. The authors say that “The analysis of the feeding microstructure of mice housed in groups and fed ad libitum suggests significant sex- and age-related contributions of meal size, meal frequency and intermeal interval to the control of net daily energy intake…” The abstract could be more specific and discuss what the sex and age effects were, similar to how they specifically described the BW results in the prior sentences. Similarly, in the next sentence, it would be informative to specifically describe the “sex-related decrease in the non-feeding time between meals.” In its current format, the results of the study are not readily discerned from the abstract.

2. The additional description of the experimental design (e.g., scrambling and group-housing etc.) is helpful to visualize the experiment. It would be even more helpful to add a supplemental figure that shows this experimental design timeline.

3. Regarding the small sample size, the authors have added one phrase to point out the small sample size in the discussion of the lack of correlation between energy expenditure and activity. However, this limitation should be discussed with regard to the entire experiment. The small sample size will affect not only affect activity-energy expenditure relationships, but also the other parameters in the experiment and this should be discussed.

Reviewer #2: The manuscript has been corrected according to my concerns. Specifically statistical findings were incorporated and interpretation of data was now restricted to present results.

7. PLOS authors have the option to publish the peer review history of their article (what does this mean?). If published, this will include your full peer review and any attached files.

Reviewer #1: No

Reviewer #2: **Yes: **Carolina Escobar

---

## [Author Response · Author response to Decision Letter 1]

20 Jan 2021

Responses to comments of Reviewer #1

1. "The abstract could be more specific and discuss what the sex and age effects were, similar to how they specifically described the BW results in the prior sentences. Similarly, in the next sentence, it would be informative to specifically describe the sex-related decrease in the non-feeding time between meals.”

Response: We have re-written a section of the abstract to indicate the sex- and age-dependent effects we have found in the feeding microstructure of mice. 

2. "The additional description of the experimental design (e.g., scrambling and group-housing etc.) is helpful to visualize the experiment. It would be even more helpful to add a supplemental figure that shows this experimental design timeline."

Response: Following the Reviewer's suggestion, a figure reflecting the experimental design time-line is now included as Supplementary Figure 1. 

3. "Regarding the small sample size, the authors have added one phrase to point out the small sample size in the discussion of the lack of correlation between energy expenditure and activity. However, this limitation should be discussed with regard to the entire experiment. The small sample size will affect not only affect activity-energy expenditure relationships, but also the other parameters in the experiment and this should be discussed."

Response: The Discussion now includes several notes of caution when interpreting our results due to the limitations imposed by a small sample size. They can be found in lines 543, 579 and 632.

---

## [Decision Letter · Decision Letter 2]

22 Jan 2021

The feeding microstructure of male and female mice

PONE-D-20-31618R2

Dear Dr. Di Fulvio,

We’re pleased to inform you that your manuscript has been judged scientifically suitable for publication and will be formally accepted for publication once it meets all outstanding technical requirements.

Kind regards,

Shin Yamazaki, Ph.D.

Section Editor

PLOS ONE

Additional Editor Comments (optional):

Reviewers' comments:

Reviewer's Responses to Questions

**Comments to the Author**

1. If the authors have adequately addressed your comments raised in a previous round of review and you feel that this manuscript is now acceptable for publication, you may indicate that here to bypass the “Comments to the Author” section, enter your conflict of interest statement in the “Confidential to Editor” section, and submit your "Accept" recommendation.

Reviewer #1: All comments have been addressed

2. Is the manuscript technically sound, and do the data support the conclusions?

Reviewer #1: Yes

3. Has the statistical analysis been performed appropriately and rigorously? 

Reviewer #1: Yes

4. Have the authors made all data underlying the findings in their manuscript fully available?

Reviewer #1: Yes

5. Is the manuscript presented in an intelligible fashion and written in standard English?

Reviewer #1: Yes

6. Review Comments to the Author

Reviewer #1: The authors have satisfactorily addressed my suggestions. I do not have any additional suggestions.

7. PLOS authors have the option to publish the peer review history of their article (what does this mean?). If published, this will include your full peer review and any attached files.

Reviewer #1: No

---

## [Editor Report · Acceptance letter]

26 Jan 2021

PONE-D-20-31618R2 

The feeding microstructure of male and female mice 

Dear Dr. Di Fulvio:

I'm pleased to inform you that your manuscript has been deemed suitable for publication in PLOS ONE. Congratulations! Your manuscript is now with our production department. 

Kind regards, 

on behalf of

Dr. Shin Yamazaki 

Section Editor

PLOS ONE